# Dynamic analysis and optimal control of stochastic information cross-dissemination and variation model with random parametric perturbations

**Sida Kang**[1], **Tianhao Liu**[2], **Hongyu Liu**[1]*, **Yuhan Hu**[3], **Xilin Hou**[1]

**1** School of Business Administration, University of Science and Technology Liaoning, Anshan, Liaoning, China, **2** Asia-Australia Business College, Liaoning University, Shenyang, Liaoning, China, **3** School of Science, University of Science and Technology Liaoning, Anshan, Liaoning, China

* Kd_lhy@163.com

**Data Availability Statement:** All relevant data are within the manuscript.

## Abstract

Information dissemination has a significant impact on social development. This paper considers that there are many stochastic factors in the social system, which will result in the phenomena of information cross-dissemination and variation. The dual-system stochastic susceptible-infectious-mutant-recovered model of information cross-dissemination and variation is derived from this problem. Afterward, the existence of the global positive solution is demonstrated, sufficient conditions for the disappearance of information and its stationary distribution are calculated, and the optimal control strategy for the stochastic model is proposed. The numerical simulation supports the results of the theoretical analysis and is compared to the parameter variation of the deterministic model. The results demonstrate that cross-dissemination of information can result in information variation and diffusion. Meanwhile, white noise has a positive effect on information dissemination, which can be improved by adjusting the perturbation parameters.

## 1 Introduction

Information generation, dissemination, and diffusion are essential to the development of human society. Generally, according to the impact of information on the society, it will show beneficial or harmful characteristics in the dissemination process. Generally, information beneficial to social development must be disseminated [1–3], whereas information detrimental to social development must be suppressed [4–6]. Therefore, it is of great importance to investigate the mode and mechanism of information dissemination.

The method and mechanism of information dissemination closely resemble those of infectious disease transmission [7, 8]. Therefore, when early scholars studied rumor spreading propagation, they usually took classical infectious disease models, such as the SI model [9], SIS model [10], and SIR model [11] as the research basis. Eventually, Daley and Kendal proposed the DK model of classical rumor spreading [12] for the first time. Subsequently, the

**Funding:** This article is supported by the Social Science Planning Fund of Liaoning Province China (No. L22AGL015).

**Competing interests:** The authors have declared that no competing interests exist.

information dissemination model drew on from the rumor-spreading model. Scholars have successively proposed the SEIR model with hesitation mechanism [13], ILSR model with individual action mechanism [14], IWSR model with self-growth mechanism [15], and SDILR model with recurrence mechanism [16].

The above-mentioned models of information dissemination typically view environmental factors as deterministic research factors, whereas in the actual social system, environmental changes are usually typically full of uncertain factors. Whether it is both the dissemination of information and the spread of infectious diseases, both are often influenced by uncertainty in environmental factors [17]. At the same time, the parameters considered in the deterministic model will also be affected by the environment. Therefore, in recent studies, researchers have increasingly focused on the impact of nondeterministic environmental factors on dissemination.

In the study on the influence of the stochastic disturbance phenomenon on the spread of infectious diseases, Zhang et al.(2018) [18] developed an SIS model with vertical dissemination for their research on the effect of stochastic disturbances on the spread of infectious diseases. They discussed how stochastic disturbance affects vertical dissemination. White noise disturbances have been found to facilitate the vertical transmission of infectious diseases. Afterward, Zhang et al.(2019) [19] improved the stochastic SIRS epidemic model with a standard incidence rate and partial immunity in their study. Researchers have discovered that random disturbances in the environment can prevent the spread of infectious diseases. Rifhat et al.(2021) [20] proposed a stochastic SIRV model with nonlinear incidence and vaccination, with the belief that stochastic fluctuations could prevent disease outbreaks. Hussain et al.(2020) [21] established a stochastic SEIR model with saturated incidence and validated the existence, uniqueness, and persistence conditions of the stochastic disturbance model caused by white noise. Bobryk (2020) [22] analyzed the influence of telegraphic noise, trichotomous noise, and bounded noise on the SIR model, as well as the influence of stochastic disturbances on the behavior of disease-free equilibrium's stability. Gutierrez et al.(2021) [23] calculated the basic reproduction numbers of SIR, SIS, and SEIR models with stochastic disturbances, and analyzed the stability of the non-deterministic models. Wang et al.(2021) [24] developed a stochastic SICA model with a standard incidence rate that provided sufficient conditions for the extermination and persistence of HIV. They discovered that the spread of the virus could be regulated by increasing the intensity of stochastic disturbance. Zhou et al.(2021) [25] developed a stochastic SIR model with nonlinear incidence and general stochastic noise, validated the existence and uniqueness of the system's stationary distribution, and determined the exact expression of the stochastic model's lognormal probability density function at another critical value. In recent years, the study of the impact of stochastic environmental disturbance on novel coronavirus transmission has also been a hot topic. He et al.(2020) [26] developed a discrete-time stochastic epidemic model with binomial distributions to study the transmission of the disease. Khan et al.(2021) [27] considered an epidemic model for coronavirus (COVID-19) with random perturbations as well as the time delay. Danane et al.(2021) [28] investigated the dynamics of a COVID-19 stochastic model with an isolation strategy. The white noise, as well as the Lévy jump perturbations, were incorporated in all compartments of the proposed model. Adak et al.(2020) [29] considered a stochastic extension of the deterministic model to capture the uncertainty or variation observed in the disease transmissibility. Chu et al.(2023) [30] constructed an $S_f M_b M_g U$ malnutrition model with random perturbations and crossover effects, and using fractional differential equations analysis deterministic-stochastic model. Rashid et al.(2023) [31] constructed an $SI_p I_q I_{qp} R_p R_q R_{qp} B$ of the co-infection of the fractional pneumonia and typhoid fever disease stochastic model with cost-effective techniques and crossover effects. Ali et al.(2023) [32] gave the dynamics analysis and simulations of stochastic COVID-19 epidemic model using Legendre spectral collocation method. In addition, Khan

et al.(2018) [33] studied on the application of Legendre spectral-collocation method to delay differential and stochastic delay differential equation.

Meanwhile, the spread of rumors will also be affected by stochastic factors. Thus, this phenomenon is of interest to some number of academics. Jia et al.(2018) [34] enhanced the stochastic SI rumor-spreading model and discovered that the obtained threshold between mean persistence and extinction is lower than that of the deterministic system. Jain et al.(2019) [35] proposed the S1S2I model with expert interaction on a homogeneous network and discovered found that noise disturbance was the cause of the continuous constant spread of rumors. Huo et al.(2020) [36] established a stochastic ISR model with white noise media coverage. The research results demonstrated that the media coverage rate was inversely proportional to the rumor's spread range. Cheng et al.(2020) [37] created the ISR model with individual activities, analyzed the effect of stochastic noise on the model's asymptotic property of the model, and presented the basic reproduction numbers of the deterministic and non-deterministic models. Li et al.(2021) [38] constructed developed a SIUR model with two distinct spread- inhibiting and attitude- adjusting mechanisms in a homogeneous social network. The results demonstrated that when the intensity of stochastic disturbance was limited, the results demonstrated that the effect of the nondeterministic model was comparable to that of the deterministic model. Zhang et al.(2022) [39] examined investigated the rumor spreading model with a general correlation function and concluded that the behavior and attitude of netizens were nondeterministic factors. The study found that government intervention and authoritative media reports would interfere with the influence Internet users' opinions, thereby preventing the spread of Internet users, thus suppressing the rumor spreading rumors. Additionally, Mena et al.(2020) and Zhou et al.(2022) analyzed the extinction and persistence of the uncertain dissemination model, and further analyzed the characteristics of dissemination by constructing the optimal control model [40, 41].

The aforementioned scholars have conducted extensive research on infectious diseases and rumor-spreading under the influence of stochastic disturbance. However, their primary focus is on the propagation of information dissemination about a single group and individual information. However, in real-world social systems, it is very common for multiple messages to transmit among different groups. At the same time, there is a certain probability that multiple pieces of information will be merged into new information after cross-dissemination among between different groups. Additionally, there are many uncertainties within the social system itself, and the dissemination of information within the social system will be affected by stochastic factors. Therefore, it is assumed that stochastic disturbances in the environment will affect influence the probability that different groups of people can access information from other groups, as well as and the probability of generating new variation information after being exposed to information from multiple groups.

In social systems multiple pieces of information sometimes coexist. The phenomenon of coexistence and intersection in the process of multiple information dissemination is similar to the spread of viruses in biology. For example, when the SARS-CoV-2 virus spread in 2020, the presence of its original strain did not cause the extinction of other strains in the body. Instead, there were multiple strains coexisting, and even new mutated strains were produced through cross transmission, such as Omicron (B.1.1.529). The phenomenon of coexisting and producing mutated strains of this virus can be analogized and applied to the spread of multiple information. Multiple similar information may have coexisting relationships, and after prolonged cross propagation, the content expressed by each information may deviate from the original information, similar to the phenomenon of multiple viruses coexisting and producing mutated viruses. This paper proposes a stochastic susceptible-infectious-mutant-recovered model that considers information cross-dissemination and variation, and then demonstrates the existence

of global positive solutions. After calculating the sufficient conditions for information disappearance and information stationary distribution, the appropriate parameters are selected as control variables. The numerical simulation validates the rationality of the proposed method is finally validated through numerical simulation.

The remaining sections of this article are organized as follows. In Section 2, the stochastic 2S2I4M2R model is developed with cross-dissemination and variation of information taken into account. The existence of global positive solutions is demonstrated in Section 3. In Section 4, sufficient conditions for the deletion of information are outlined. Section 5 provides sufficient conditions for stationary information distribution. Section 6 proposes the existence and strategies for optimal control of information cross-distribution and variation. In Section 7, numerical simulation is used to analyze the impact of stochastic disturbance strength on information cross-dissemination and variation. The last Section draws the conclusion.

## 2 The model

In this paper, an open virtual community is considered in which the population size varies over time t, and the total population size is represented by $N(t)$. All populations can be classified into the following eight categories: (1) the easy-to-adopt populations in the two groups who have not been exposed to information but are receptive to adopting information, represented by $S_1(t)$ and $S_2(t)$. (2) The group who are simultaneously exposed to both types of information but choose to disseminate the first type, represented by $I_1(t)$. (3) The group who are simultaneously exposed to both types of information but choose to spread the second type, represented by $I_2(t)$. (4) The disseminators who are simultaneously exposed to both types of information, but disseminate the first type of information and ultimately choose the variation group who strongly believe the first type of information, represented by $M_1(t)$. (5) The disseminators who are exposed to both kinds of information but disseminate the first kind of information will finally select the variation population that integrate the two kinds of information, represented by $M_2(t)$. (6) The disseminators, who are simultaneously exposed to both types of information but disseminate the second type, will ultimately select the variation population that integrates both types of information, represented by $M_3(t)$. (7) The disseminators who are simultaneously exposed to both types of information but choose to disseminate the second type will eventually choose the variation population who strongly believe the second type of information, as represented by $M_4(t)$. (8) In the two groups, the escaping group, who are not interested in the two kinds of information and the variation of the two kinds of information, are represented by $R_1(t)$ and $R_2(t)$.

This paper constructs a model to reflect the phenomenon of information cross-dissemination and variation. The model flow diagram is given in Fig 1.

Parameters in the model can be interpreted as follows:

- The number of individuals in a social system generally changes over time. Therefore, in this paper, $B$ is defined as the number of people moving within the social system, whereas $\mu$ is defined as the rate of people leaving the social system due to force majeure.

- When two types of information begin to spread between two groups, the easy-to-adopt group will have a certain probability of contacting the population that is spreading the information. The easy-to-adopt group within the same group has priority to contact the information disseminators of their own group, followed by a probability to contact the information disseminators of another group. Therefore, the probabilities of the easy-to-adopt population in the two groups contacting the information disseminators of their own group are defined

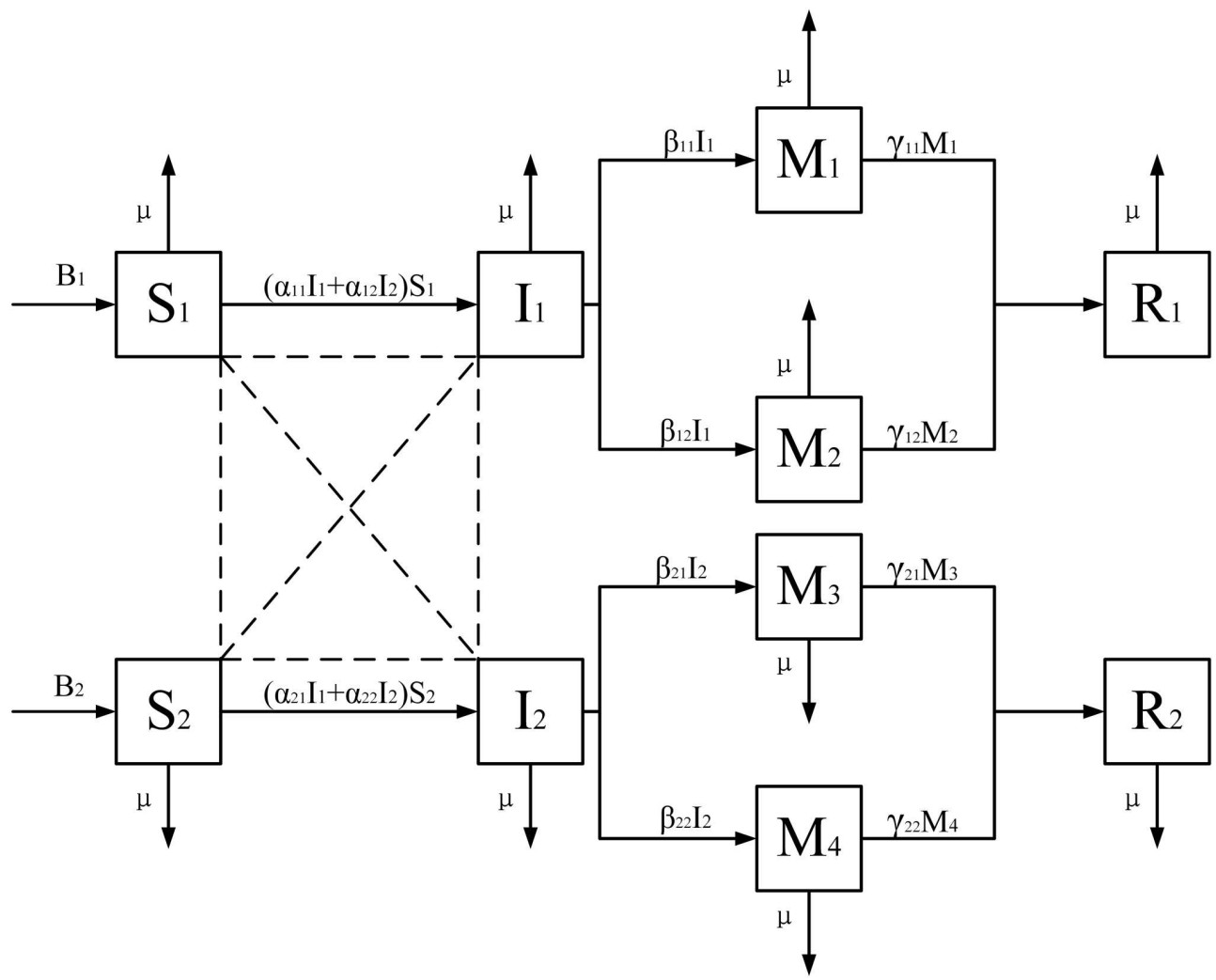

**Fig 1. The flow diagram of the model.**

as $\alpha_{11}$ and $\alpha_{22}$, respectively, and the probabilities of contacting the information disseminators of another group are defined as $\alpha_{12}$ and $\alpha_{21}$, respectively.

- When the disseminators who receive two kinds of information at the same time integrate the information, there will be a certain probability to produce new variation information. When the disseminators of the two types of information strongly believe the original information, they will mutate into the $M_1$ and $M_4$ groups, with a probability of $\beta_{11}$ and $\beta_{22}$. When the disseminators of two types of information choose to combine the information, they will mutate into $M_2$ and $M_3$ groups with probabilities $\beta_{12}$ and $\beta_{21}$, where the fusion information generated by the former is dominated by the first information and supplemented by the second, and the latter is the opposite.

- Information may be eliminated after a period of dissemination, as it is generally effective. With the probabilities $\gamma_{11}$, $\gamma_{12}$, $\gamma_{21}$, and $\gamma_{22}$, the mutation group will switch to the escaping group.

The system dynamics equations are described as follows:

$$
\begin{cases}
\dfrac{dS_1}{dt} &= B_1 - \alpha_{11}S_1I_1 - \alpha_{12}S_1I_2 - \mu S_1, \\[2mm]
\dfrac{dI_1}{dt} &= \alpha_{11}S_1I_1 + \alpha_{12}S_1I_2 - (\beta_{11} + \beta_{12} + \mu)I_1, \\[2mm]
\dfrac{dM_1}{dt} &= \beta_{11}I_1 - (\gamma_{11} + \mu)M_1, \\[2mm]
\dfrac{dM_2}{dt} &= \beta_{12}I_1 - (\gamma_{12} + \mu)M_2, \\[2mm]
\dfrac{dR_1}{dt} &= \gamma_{11}M_1 + \gamma_{12}M_2 - \mu R_1. \\[2mm]
\dfrac{dS_2}{dt} &= B_2 - \alpha_{21}S_2I_1 - \alpha_{22}S_2I_2 - \mu S_2, \\[2mm]
\dfrac{dI_2}{dt} &= \alpha_{21}S_2I_1 + \alpha_{22}S_2I_2 - (\beta_{21} + \beta_{22} + \mu)I_2, \\[2mm]
\dfrac{dM_3}{dt} &= \beta_{21}I_2 - (\gamma_{21} + \mu)M_3, \\[2mm]
\dfrac{dM_4}{dt} &= \beta_{22}I_2 - (\gamma_{22} + \mu)M_4, \\[2mm]
\dfrac{dR_2}{dt} &= \gamma_{21}M_3 + \gamma_{22}M_4 - \mu R_2.
\end{cases}
\tag{1}
$$

Where:

$$
\begin{aligned}
&B_1 > 0, B_2 > 0, \mu > 0, \gamma_{11} > 0, \gamma_{12} > 0, \gamma_{21} > 0, \gamma_{22} > 0, \\
&\alpha_{11} \in (0,1], \alpha_{12} \in (0,1], \alpha_{21} \in (0,1], \alpha_{22} \in (0,1], \\
&\beta_{11} \in (0,1], \beta_{12} \in (0,1], \beta_{21} \in (0,1], \beta_{22} \in (0,1],
\end{aligned}
\tag{2}
$$

and

$$
S_1(t) + I_1(t) + M_1(t) + M_2(t) + R_1(t) + S_2(t) + I_2(t) + M_3(t) + M_4(t) + R_2(t) = N(t). \tag{3}
$$

Furthermore, there are many uncertain environmental factors in the social system, which can be referred to as environmental noise. It is unscientific to build a model of information dissemination that disregards stochastic environmental noise disturbance. Adding environmental noise to the deterministic model makes information dissemination in social systems more realistic. There are three traditional methods for incorporating stochastic factors into the model of dispersion: (1) White Gaussian noise is added to the deterministic model to account for parameter perturbation [42]. (2) Random perturbation will surround the deterministic model at the positive equilibrium point [43]. (3) The system will transition between regimes in accordance with the probability law of the Markov chain [44]. Social system stochastic factors will influence the contact rate between the easy-to-adopt population and the information communicator and the mutation rate of the information disseminator. In this paper, the stochastic disturbances of $\alpha_{11}$, $\alpha_{12}$, $\alpha_{22}$, and $\alpha_{21}$ as well as $\beta_{11}$, $\beta_{12}$, $\beta_{22}$, and $\beta_{21}$ are characterized using

Gaussian white noise. These parameters for stochastic disturbances can be expressed as:

$$
\begin{aligned}
&\alpha_{11} \to \alpha_{11} + \sigma_1 \dot{W}_1(t), \alpha_{12} \to \alpha_{12} + \sigma_2 \dot{W}_2(t), \\
&\alpha_{21} \to \alpha_{21} + \sigma_3 \dot{W}_3(t), \alpha_{22} \to \alpha_{22} + \sigma_4 \dot{W}_4(t), \\
&\beta_{11} \to \beta_{11} + \sigma_5 \dot{W}_5(t), \beta_{12} \to \beta_{12} + \sigma_6 \dot{W}_6(t), \\
&\beta_{21} \to \beta_{21} + \sigma_7 \dot{W}_7(t), \beta_{22} \to \beta_{22} + \sigma_8 \dot{W}_8(t).
\end{aligned}
\tag{4}
$$

Here, $W_i(i = 1 \sim 8)$ are independent standard Brownian motions and $\sigma_i^2 > 0(i = 1 \sim 8)$ represent the intensities of $W_i(i = 1 \sim 8)$, respectively.

Then, the stochastic disturbance parameters are introduced into the deterministic model to establish the stochastic 2S2I4M2R model driven by white Gaussian noise. The stochastic model can be expressed as:

$$
\begin{cases}
dS_1(t) &= (B_1 - \alpha_{11}S_1I_1 - \alpha_{12}S_1I_2 - \mu S_1)dt - \sigma_1 S_1 I_1 dW_1(t) \\
&- \sigma_2 S_1 I_2 dW_2(t), \\
dI_1(t) &= (\alpha_{11}S_1I_1 + \alpha_{12}S_1I_2 - (\beta_{11} + \beta_{12} + \mu)I_1)dt + \sigma_1 S_1 I_1 dW_1(t) \\
&+ \sigma_2 S_1 I_2 dW_2(t) - \sigma_5 I_1 dW_5(t) - \sigma_6 I_1 dW_6(t), \\
dM_1(t) &= (\beta_{11}I_1 - (\gamma_{11} + \mu)M_1)dt + \sigma_5 I_1 dW_5(t), \\
dM_2(t) &= (\beta_{12}I_1 - (\gamma_{12} + \mu)M_2)dt + \sigma_6 I_1 dW_6(t). \\
dS_2(t) &= (B_2 - \alpha_{21}S_2I_1 - \alpha_{22}S_2I_2 - \mu S_2)dt - \sigma_3 S_2 I_1 dW_3(t) \\
&- \sigma_4 S_2 I_2 dW_4(t), \\
dI_2(t) &= (\alpha_{21}S_2I_1 + \alpha_{22}S_2I_2 - (\beta_{21} + \beta_{22} + \mu)I_2)dt + \sigma_3 S_2 I_1 dW_3(t) \\
&+ \sigma_4 S_2 I_2 dW_4(t) - \sigma_7 I_2 dW_7(t) - \sigma_8 I_2 dW_8(t), \\
dM_3(t) &= (\beta_{21}I_2 - (\gamma_{21} + \mu)M_3)dt + \sigma_7 I_2 dW_7(t), \\
dM_4(t) &= (\beta_{22}I_2 - (\gamma_{22} + \mu)M_4)dt + \sigma_8 I_2 dW_8(t).
\end{cases}
\tag{5}
$$

## 3 Existence of the global and positive solution

In the rest of this paper, let $(\Omega, \mathscr{F}, \{\mathscr{F}_t\}_{t \geq 0}, P)$ be a complete probability space with a filtration $\{\mathscr{F}_t\}_{t \geq 0}$ satisfying the usual conditions. And while $\mathscr{F}_0$ contains all $P - null$ sets, it is increasing and right continuous [45]. It also can be denoted as:

$$
\mathbb{R}_+^8 = \{(x_1, x_2, x_3, x_4, x_5, x_6, x_7, x_8)|x_i > 0, i = 1, 2, 3, 4, 5, 6, 7, 8\}.
$$

Whether the global solution is existence that the basis of analyzing the dynamic behavior of stochastic system (5). At the same time, according to the actual situation, it can be required a positive value for the dynamic model of information transmission. The stochastic system (5) can be proved global and positive by Theorem 2.1.

**Theorem 2.1** *The existence of a unique positive solution*
$(S_1(t), I_1(t), M_1(t), M_2(t), S_2(t), I_2(t), M_3(t), M_4(t)) \in \mathbb{R}_+^8$ *of stochastic system* (5) *is satisfied any given initial value* $(S_1(0), I_1(0), M_1(0), M_2(0), S_2(0), I_2(0), M_3(0), M_4(0)) \in \mathbb{R}_+^8$. *The probability of the solution is 1 and remains in* $\mathbb{R}_+^8$.

**Proof of Theorem 2.1** *The existence of a unique local positive solution*
$(S_1(t), I_1(t), M_1(t), M_2(t), S_2(t), I_2(t), M_3(t), M_4(t)) \in \mathbb{R}_+^8$ *of stochastic system* (5) *on* $t \in [0, \tau_e)$, *which is based on the coefficients of system* (1) *are locally Lipschitz continuous of any given initial*

*value* $(S_1(0), I_1(0), M_1(0), M_2(0), S_2(0), I_2(0), M_3(0), M_4(0)) \in \mathbb{R}_+^8$. $\tau_e$ *is the explosion time* [46]. *It is need to have that* $\tau_e = \infty$ *a.s. to show this solution globally. The stopping time* $\tau^+$ *can be defined by:*

$$\tau^+ = \inf\{t \in [0, \tau_e) : S_1(t) \le 0 \text{ or } I_1(t) \le 0 \text{ or } M_1(t) \le 0 \text{ or } M_2(t) \le 0$$

$$\text{or } S_2(t) \le 0 \text{ or } I_2(t) \le 0 \text{ or } M_3(t) \le 0 \text{ or } M_4(t) \le 0\}.$$

*Let set* $\inf \emptyset = \infty$ ($\emptyset$ *denotes the empty set). It is easy to get* $\tau^+ \le \tau_e$. *So if* $\tau^+ = \infty$ *a.s. is proved, then* $\tau_e = \infty$ *and* $(S_1(t), I_1(t), M_1(t), M_2(t), S_2(t), I_2(t), M_3(t), M_4(t)) \in \mathbb{R}_+^8$ *a.s. for all* $t \ge 0$. *Assume that* $\tau^+ < \infty$, *then* $T > 0$ *is existence such that* $P(\tau^+ < T) > 0$. *Define* $C^2$ *function* $V$: $\mathbb{R}_+^8 \to \mathbb{R}_+^8$ *by* $V(X) = \ln S_1 S_2 I_1 I_2 M_1 M_2 M_3 M_4$. *Let using Itô's formula to calculate the differential of* $V$ *along the solution trajectories of stochastic system* (5). *We get, for* $\omega \in (\tau^+ < T)$, *and for all* $t \in [0, \tau_e)$,

$$
\begin{aligned}
dV(X(t)) =\ & \left(\frac{B_1}{S_1} - \alpha_{11}I_1 - \alpha_{12}I_2 - \mu - \frac{1}{2}\sigma_1^2 I_1^2 - \frac{1}{2}\sigma_2^2 I_2^2\right)dt + \left[\alpha_{11}S_1 + \frac{\alpha_{12}S_1 I_2}{I_1}\right. \\
& - \left.(\beta_{11} + \beta_{12} + \mu) - \frac{1}{2}\sigma_1^2 S_1^2 - \frac{1}{2}\sigma_2^2\left(\frac{S_1 I_2}{I_1}\right)^2 - \frac{1}{2}\sigma_5^2 - \frac{1}{2}\sigma_6^2\right]dt + \left[\frac{\beta_{11}I_1}{M_1}\right. \\
& - \left.(\gamma_{11} + \mu) - \frac{1}{2}\sigma_5^2\left(\frac{I_1}{M_1}\right)^2\right]dt + \left[\frac{\beta_{12}I_1}{M_2} - (\gamma_{12} + \mu) - \frac{1}{2}\sigma_6^2\left(\frac{I_1}{M_2}\right)^2\right]dt \\
& + \left(\frac{B_2}{S_2} - \alpha_{21}I_1 - \alpha_{22}I_2 - \mu - \frac{1}{2}\sigma_3^2 I_1^2 - \frac{1}{2}\sigma_4^2 I_2^2\right)dt + \left[\frac{\alpha_{21}S_2 I_1}{I_2} + \alpha_{22}S_2\right. \\
& - \left.(\beta_{21} + \beta_{22} + \mu) - \frac{1}{2}\sigma_3^2\left(\frac{S_2 I_1}{I_2}\right)^2 - \frac{1}{2}\sigma_4^2 S_2^2 - \frac{1}{2}\sigma_7^2 - \frac{1}{2}\sigma_8^2\right]dt + \left[\frac{\beta_{21}I_2}{M_3}\right. \quad (6) \\
& - \left.(\gamma_{21} + \mu) - \frac{1}{2}\sigma_7^2\left(\frac{I_2}{M_3}\right)^2\right]dt + \left[\frac{\beta_{22}I_2}{M_4} - (\gamma_{22} + \mu) - \frac{1}{2}\sigma_8^2\left(\frac{I_2}{M_4}\right)^2\right]dt \\
& - \sigma_1 I_1 dW_1 - \sigma_2 I_2 dW_2 + \sigma_1 S_1 dW_1 + \frac{\sigma_2 S_1 I_2}{I_1}dW_2 - \sigma_5 dW_5 - \sigma_6 dW_6 \\
& + \frac{\sigma_5 I_1}{M_1}dW_5 + \frac{\sigma_6 I_1}{M_2}dW_6 - \sigma_3 I_1 dW_3 - \sigma_4 I_2 dW_4 + \frac{\sigma_3 S_2 I_1}{I_2}dW_3 \\
& + \sigma_4 S_2 dW_4 - \sigma_7 dW_7 - \sigma_8 dW_8 + \frac{\sigma_7 I_2}{M_3}dW_7 + \frac{\sigma_8 I_2}{M_4}dW_8.
\end{aligned}
$$

*Positivity of* $X(t)$ *implies that*

$$
\begin{aligned}
dV(X(t)) \ge\ & L(S_1, I_1, M_1, M_2, S_2, I_2, M_3, M_4)dt - \sigma_1(I_1 - S_1)dW_1 \\
& - \sigma_2(I_2 - S_1 I_2)dW_2 - \sigma_3\left(I_1 - \tfrac{S_2 I_1}{I_2}\right)dW_3 - \sigma_4(I_2 - S_2)dW_4 \\
& - \sigma_5\left(1 - \tfrac{I_1}{M_1}\right)dW_5 - \sigma_6\left(1 - \tfrac{I_1}{M_2}\right)dW_6 - \sigma_7\left(1 - \tfrac{I_2}{M_3}\right)dW_7 \quad (7) \\
& - \sigma_8\left(1 - \tfrac{I_2}{M_4}\right)dW_8,
\end{aligned}
$$

*where*

$$
\begin{aligned}
L(S_1, I_1, M_1, M_2, S_2, I_2, M_3, M_4) =\ & -\mu - (\beta_{11} + \beta_{12} + \mu) - (\gamma_{11} + \mu) - (\gamma_{12} + \mu) - \mu \\
& - (\beta_{21} + \beta_{22} + \mu) - (\gamma_{21} + \mu) - (\gamma_{22} + \mu) - \tfrac{1}{2}\sigma_1^2 I_1^2 \\
& - \tfrac{1}{2}\sigma_1^2 S_1^2 - \tfrac{1}{2}\sigma_2^2 I_2^2 - \tfrac{1}{2}\sigma_2^2 \left(\tfrac{S_1 I_2}{I_1}\right)^2 - \tfrac{1}{2}\sigma_3^2 I_1^2 \\
& - \tfrac{1}{2}\sigma_3^2 \left(\tfrac{S_2 I_1}{I_2}\right)^2 - \tfrac{1}{2}\sigma_4^2 I_2^2 - \tfrac{1}{2}\sigma_4^2 S_2^2 - \tfrac{1}{2}\sigma_5^2 \\
& - \tfrac{1}{2}\sigma_5^2 \left(\tfrac{I_1}{M_1}\right)^2 - \tfrac{1}{2}\sigma_6^2 - \tfrac{1}{2}\sigma_6^2 \left(\tfrac{I_1}{M_2}\right)^2 - \tfrac{1}{2}\sigma_7^2 \\
& - \tfrac{1}{2}\sigma_7^2 \left(\tfrac{I_2}{M_3}\right)^2 - \tfrac{1}{2}\sigma_8^2 - \tfrac{1}{2}\sigma_8^2 \left(\tfrac{I_2}{M_4}\right)^2.
\end{aligned}
\tag{8}
$$

*So we have*

$$
\begin{aligned}
V(X(t)) \geq\ & V(X_0) + \int_0^t L(S_1(u), I_1(u), M_1(u), M_2(u), S_2(u), I_2(u), M_3(u), M_4(u))\,du \\
& - \int_0^t \sigma_1(I_1(u) - S_1(u))\,dW_1(u) - \int_0^t \sigma_2(I_2(u) - S_1(u)I_2(u))\,dW_2(u) \\
& - \int_0^t \sigma_3\left(I_1(u) - \frac{S_2(u)I_1(u)}{I_2(u)}\right)dW_3(u) - \int_0^t \sigma_4(I_2(u) - S_2(u))\,dW_4(u) \\
& - \int_0^t \sigma_5\left(1 - \frac{I_1(u)}{M_1(u)}\right)dW_5(u) - \int_0^t \sigma_6\left(1 - \frac{I_1(u)}{M_2(u)}\right)dW_6(u) \\
& - \int_0^t \sigma_7\left(1 - \frac{I_2(u)}{M_3(u)}\right)dW_7(u) - \int_0^t \sigma_8\left(1 - \frac{I_2(u)}{M_4(u)}\right)dW_8(u).
\end{aligned}
\tag{9}
$$

*Note that some components of $X(\tau^+)$ equal 0. Thereby*

$$
\lim_{t \to \tau^+} V(X(t)) = -\infty.
$$

*Letting $t \to \tau^+$ in system* (9), *one have*

$$
\begin{aligned}
& V(X_0) + \int_0^{\tau^+} L(S_1(u), I_1(u), M_1(u), M_2(u), S_2(u), I_2(u), M_3(u), M_4(u))\,du \\
& - \int_0^{\tau^+} \sigma_1(I_1(u) - S_1(u))\,dW_1(u) - \int_0^{\tau^+} \sigma_2(I_2(u) - S_1(u)I_2(u))\,dW_2(u) \\
& - \int_0^{\tau^+} \sigma_3\left(I_1(u) - \tfrac{S_2(u)I_1(u)}{I_2(u)}\right)dW_3(u) - \int_0^{\tau^+} \sigma_4(I_2(u) - S_2(u))\,dW_4(u) \\
& - \int_0^{\tau^+} \sigma_5\left(1 - \tfrac{I_1(u)}{M_1(u)}\right)dW_5(u) - \int_0^{\tau^+} \sigma_6\left(1 - \tfrac{I_1(u)}{M_2(u)}\right)dW_6(u) \\
& - \int_0^{\tau^+} \sigma_7\left(1 - \tfrac{I_2(u)}{M_3(u)}\right)dW_7(u) - \int_0^{\tau^+} \sigma_8\left(1 - \tfrac{I_2(u)}{M_4(u)}\right)dW_8(u) > -\infty,
\end{aligned}
\tag{10}
$$

*it contradicts the assumption. Thus, $\tau^+ = \infty$.*

## 4 Disappearance of the information

Theorem 3.1 gives the condition for the disappearance of the information. The condition is expressed by intensities of noises and parameters of system (1).

**Theorem 3.1** *For any given initial value*
$(S_1(0), I_1(0), M_1(0), M_2(0), S_2(0), I_2(0), M_3(0), M_4(0)) \in \mathbb{R}_+^8$, $\limsup\limits_{t \to \infty} \frac{\ln I_1(t)}{t} \leq$

$G(\sigma_1^2, \sigma_2^2, \sigma_5^2, \sigma_6^2)$ and $\limsup\limits_{t\to\infty} \frac{\ln I_2(t)}{t} \le G(\sigma_3^2, \sigma_4^2, \sigma_7^2, \sigma_8^2)$ holds a.s.. Further, $G(\sigma_1^2, \sigma_2^2, \sigma_5^2, \sigma_6^2) < 0$ and $G(\sigma_3^2, \sigma_4^2, \sigma_7^2, \sigma_8^2) < 0$, then $I_1(t)$ and $I_2(t)$ tend to 0 exponentially a.s., where

$G(\sigma_1^2, \sigma_2^2, \sigma_5^2, \sigma_6^2) = \frac{\alpha_{11}^2}{2\sigma_1^2} + \frac{\alpha_{12}^2}{2\sigma_2^2} - \left(\beta_{11} + \beta_{12} + \mu + \frac{1}{2}\sigma_5^2 + \frac{1}{2}\sigma_6^2\right)$ and

$G(\sigma_3^2, \sigma_4^2, \sigma_7^2, \sigma_8^2) = \frac{\alpha_{21}^2}{2\sigma_3^2} + \frac{\alpha_{22}^2}{2\sigma_4^2} - \left(\beta_{21} + \beta_{22} + \mu + \frac{1}{2}\sigma_7^2 + \frac{1}{2}\sigma_8^2\right).$

**Proof of Theorem 3.1** *Using Itô's formula and by the stochastic system* (5), *$d\ln I_1(t)$ and $d\ln I_2(t)$ can be written as:*

$$
\begin{aligned}
d\ln I_1(t) &= [\alpha_{11}S_1 + \tfrac{\alpha_{12}S_1 I_2}{I_1} - (\beta_{11} + \beta_{12} + \mu) - \tfrac{1}{2}\sigma_1^2 S_1^2 - \tfrac{1}{2}\sigma_2^2\left(\tfrac{S_1 I_2}{I_1}\right)^2 \\
&\quad - \tfrac{1}{2}\sigma_5^2 - \tfrac{1}{2}\sigma_6^2]dt + \sigma_1 S_1 dW_1 + \tfrac{\sigma_2 S_1 I_2}{I_1}dW_2 - \sigma_5 dW_5 - \sigma_6 dW_6,
\end{aligned}
\tag{11}
$$

*and*

$$
\begin{aligned}
d\ln I_2(t) &= [\tfrac{\alpha_{21}S_2 I_1}{I_2} + \alpha_{22}S_2 - (\beta_{21} + \beta_{22} + \mu) - \tfrac{1}{2}\sigma_3^2\left(\tfrac{S_2 I_1}{I_2}\right)^2 - \tfrac{1}{2}\sigma_4^2 S_2^2 \\
&\quad - \tfrac{1}{2}\sigma_7^2 - \tfrac{1}{2}\sigma_8^2]dt + \tfrac{\sigma_3 S_2 I_1}{I_2}dW_3 + \sigma_4 S_2 dW_4 - \sigma_7 dW_7 - \sigma_8 dW_8.
\end{aligned}
\tag{12}
$$

*Thus, $\ln I_1(t)$ and $\ln I_2(t)$ can be denoted as:*

$$
\begin{aligned}
\ln I_1(t) &= \ln I_1(0) + \int_o^t \left[ \begin{array}{l} \alpha_{11}S_1(u) + \dfrac{\alpha_{12}S_1(u)I_2(u)}{I_1(u)} - (\beta_{11} + \beta_{12} + \mu) \\[2mm] -\dfrac{1}{2}\sigma_1^2 S_1^2(u) - \dfrac{1}{2}\sigma_2^2\left(\dfrac{S_1(u)I_2(u)}{I_1(u)}\right)^2 - \dfrac{1}{2}\sigma_5^2 - \dfrac{1}{2}\sigma_6^2 \end{array} \right] du \\[2mm]
&\quad + \int_0^t \sigma_1 S_1(u)dW_1(u) + \int_0^t \dfrac{\sigma_2 S_1(u)I_2(u)}{I_1(u)}dW_2(u) - \sigma_5 dW_5(t) - \sigma_6 dW_6(t),
\end{aligned}
\tag{13}
$$

*and*

$$
\begin{aligned}
\ln I_2(t) &= \ln I_2(0) + \int_0^t \left[ \begin{array}{l} \frac{\alpha_{21}S_2(u)I_1(u)}{I_2(u)} + \alpha_{22}S_2(u) - (\beta_{21} + \beta_{22} + \mu) \\[2mm] -\frac{1}{2}\sigma_3^2\left(\frac{S_2(u)I_1(u)}{I_2(u)}\right)^2 - \frac{1}{2}\sigma_4^2 S_2^2(u) - \frac{1}{2}\sigma_7^2 - \frac{1}{2}\sigma_8^2 \end{array} \right] du \\[2mm]
&\quad + \int_0^t \dfrac{\sigma_3 S_2(u)I_1(u)}{I_2(u)}dW_3(u) + \int_0^t \sigma_4 S_2(u)dW_4(u) - \sigma_7 dW_7(t) - \sigma_8 dW_8(t).
\end{aligned}
\tag{14}
$$

*Denote*

$$
\Phi_1(t) = \int_0^t \sigma_1 S_1(u)dW_1(u), \Phi_2(t) = \int_0^t \sigma_2(S_1(u)I_1(u)/I_2(u))dW_2(u),
$$

$$
\Phi_3(t) = \int_0^t \sigma_3(S_2(u)I_1(u)/I_2(u))dW_3(u), \Phi_4(t) = \int_0^t \sigma_4 S_2(u)dW_4(u).
\tag{15}
$$

*$\Phi_1(t), \Phi_2(t), \Phi_3(t), \Phi_4(t)$ are continuous local martingale. The quadratic variation of $\Phi_1(t)$, $\Phi_2(t), \Phi_3(t), \Phi_4(t)$ can be denoted as:*

$$
\langle \Phi_1(t) \rangle = \sigma_1^2 \int_0^t S_1^2(u)du, \langle \Phi_2(t) \rangle = \sigma_2^2 \int_0^t (S_1^2(u)I_1^2(u)/I_2^2(u))du,
$$

$$
\langle \Phi_3(t) \rangle = \sigma_3^2 \int_0^t (S_2^2(u)I_1^2(u)/I_2^2(u))du, \langle \Phi_4(t) \rangle = \sigma_4^2 \int_0^t S_2^2(u)du.
\tag{16}
$$

*By Lemma 2.2 in* [47] *and Lemma 3.1 in* [45], *one gets*

$$P\left\{\sup_{0\leq t\leq k}\left[\Phi(t)-\frac{c}{2}\langle\Phi(t)\rangle\right]>\frac{2}{c}\ln k\right\}\leq k^{-\frac{2}{c}},$$

*where* $0<c<1$, $k$ *is a random integer. Using Borel-Cantelli lemma, it is easy to know that the random integer* $k_0(\omega)$ *exists such that for* $k>k_0$ *for almost all* $\omega\in\Omega$,
$\sup_{0\leq t\leq k}\left[\Phi(t)-\frac{c}{2}\langle\Phi(t)\rangle\right]\leq\frac{2}{c}$. *Therefore, for all* $t\in[0,k]$, *one have*

$$\int_0^t\sigma_1 S_1(u)dW_1(u)\leq\frac{c}{2}\sigma_1^2\int_0^t S_1^2(u)du+\frac{2}{c}\ln k,\tag{17}$$

$$\int_0^t\sigma_2\frac{S_1(u)I_1(u)}{I_2(u)}dW_2(u)\leq\frac{c}{2}\sigma_2^2\int_0^t\frac{S_1^2(u)I_1^2(u)}{I_2^2(u)}du+\frac{2}{c}\ln k,\tag{18}$$

$$\int_0^t\sigma_3\frac{S_2(u)I_1(u)}{I_2(u)}dW_3(u)\leq\frac{c}{2}\sigma_3^2\int_0^t\frac{S_2^2(u)I_1^2(u)}{I_2^2(u)}du+\frac{2}{c}\ln k,\tag{19}$$

$$\int_0^t\sigma_4 S_2(u)dW_4(u)\leq\frac{c}{2}\sigma_4^2\int_0^t S_2^2(u)du+\frac{2}{c}\ln k.\tag{20}$$

*Then, it can be obtained that*

$$\begin{aligned}\ln I_1(t)&\leq\ln I_1(0)+\int_0^t\left[\begin{array}{l}\alpha_{11}S_1(u)+\frac{\alpha_{12}S_1(u)I_2(u)}{I_1(u)}-(\beta_{11}+\beta_{12}+\mu)-\frac{1}{2}\sigma_5^2\\-\frac{1}{2}\sigma_6^2-\frac{1}{2}(1-c)\sigma_1^2 S_1^2(u)-\frac{1}{2}(1-c)\sigma_2^2\left(\frac{S_1(u)I_2(u)}{I_1(u)}\right)^2\end{array}\right]du\\&+\frac{2}{c}\ln k+\frac{2}{c}\ln k-\sigma_5 W_5(t)-\sigma_6 W_6(t),\end{aligned}\tag{21}$$

*Noting that*

$$\alpha_{11}S_1(u)-\frac{1}{2}(1-c)\sigma_1^2 S_1^2(u)\leq\frac{\alpha_{11}^2}{2(1-c)\sigma_1^2},\tag{22}$$

*and*

$$\frac{\alpha_{12}S_1(u)I_2(u)}{I_1(u)}-\frac{1}{2}(1-c)\sigma_2^2\left(\frac{S_1(u)I_2(u)}{I_1(u)}\right)^2\leq\frac{\alpha_{12}^2}{2(1-c)\sigma_2^2}.\tag{23}$$

*Similarly*

$$\begin{aligned}\ln I_2(t)&\leq\ln I_2(0)+\int_0^t\left[\begin{array}{l}\int_0^t[\frac{\alpha_{21}S_2(u)I_1(u)}{I_2(u)}+\alpha_{22}S_2(u)-(\beta_{21}+\beta_{22}+\mu)-\frac{1}{2}\sigma_7^2\\-\frac{1}{2}\sigma_8^2-\frac{1}{2}(1-c)\sigma_3^2\left(\frac{S_2(u)I_1(u)}{I_2(u)}\right)^2-\frac{1}{2}(1-c)\sigma_4^2 S_2^2(u)\end{array}\right]du\\&+\frac{2}{c}\ln k+\frac{2}{c}\ln k-\sigma_7 W_7(t)-\sigma_8 W_8(t).\end{aligned}\tag{24}$$

*Noting that*

$$\frac{\alpha_{21}S_2(u)I_1(u)}{I_2(u)} - \frac{1}{2}(1-c)\sigma_3^2\left(\frac{S_2(u)I_1(u)}{I_2(u)}\right)^2 \le \frac{\alpha_{21}^2}{2(1-c)\sigma_3^2},\tag{25}$$

*and*

$$\alpha_{22}S_2(u) - \frac{1}{2}(1-c)\sigma_4^2 S_2^2(u) \le \frac{\alpha_{22}^2}{2(1-c)\sigma_4^2}.\tag{26}$$

*Substituting* Eqs (22) *and* (23) *into* (21), $\ln I_1(t)$ *can be written as:*

$$
\begin{aligned}
\ln I_1(t) \quad \le \quad & \ln I_1(0) + \int_0^t\left[\frac{\alpha_{11}^2}{2(1-c)\sigma_1^2} + \frac{\alpha_{12}^2}{2(1-c)\sigma_2^2} - \left(\beta_{11}+\beta_{12}+\mu+\frac{1}{2}\sigma_5^2+\frac{1}{2}\sigma_6^2\right)\right]du \\
+ \quad & \frac{2}{c}\ln k + \frac{2}{c}\ln k - \sigma_5 W_5(t) - \sigma_6 W_6(t) \\
= \quad & \ln I_1(0) + \left[\frac{\alpha_{11}^2}{2(1-c)\sigma_1^2} + \frac{\alpha_{12}^2}{2(1-c)\sigma_2^2} - \left(\beta_{11}+\beta_{12}+\mu+\frac{1}{2}\sigma_5^2+\frac{1}{2}\sigma_6^2\right)\right]t \\
+ \quad & \frac{2}{c}\ln k + \frac{2}{c}\ln k - \sigma_5 W_5(t) - \sigma_6 W_6(t).
\end{aligned}\tag{27}
$$

*Similarly, substituting* Eqs (25) *and* (26) *into* (24), $\ln I_2(t)$ *can be written as:*

$$
\begin{aligned}
\ln I_2(t) \quad \le \quad & \ln I_2(0) + \int_0^t[\frac{\alpha_{21}^2}{2(1-c)\sigma_3^2} + \frac{\alpha_{22}^2}{2(1-c)\sigma_4^2} - (\beta_{21}+\beta_{22}+\mu+\frac{1}{2}\sigma_7^2+\frac{1}{2}\sigma_8^2)]du \\
+ \quad & \frac{2}{c}\ln k + \frac{2}{c}\ln k - \sigma_7 W_7(t) - \sigma_8 W_8(t) \\
= \quad & \ln I_2(0) + \left[\frac{\alpha_{21}^2}{2(1-c)\sigma_3^2} + \frac{\alpha_{22}^2}{2(1-c)\sigma_4^2} - \left(\beta_{21}+\beta_{22}+\mu+\frac{1}{2}\sigma_7^2+\frac{1}{2}\sigma_8^2\right)\right]t \\
+ \quad & \frac{2}{c}\ln k + \frac{2}{c}\ln k - \sigma_7 W_7(t) - \sigma_8 W_8(t).
\end{aligned}\tag{28}
$$

*Hence, for $k-1 \le t \le k$, $\frac{\ln I_1(t)}{t}$ and $\frac{\ln I_2(t)}{t}$ can be obtained as:*

$$
\begin{aligned}
\frac{\ln I_1(t)}{t} \quad \le \quad & \frac{\ln I_1(0)}{t} + \frac{\alpha_{11}^2}{2(1-c)\sigma_1^2} + \frac{\alpha_{12}^2}{2(1-c)\sigma_2^2} - \left(\beta_{11}+\beta_{12}+\mu+\frac{1}{2}\sigma_5^2+\frac{1}{2}\sigma_6^2\right) \\
+ \quad & \frac{2}{c}\frac{\ln k}{k-1} + \frac{2}{c}\frac{\ln k}{k-1} - \sigma_5\frac{W_5(t)}{t} - \sigma_6\frac{W_6(t)}{t},
\end{aligned}\tag{29}
$$

*and*

$$
\begin{aligned}
\frac{\ln I_2(t)}{t} \quad \le \quad & \frac{\ln I_2(0)}{t} + \frac{\alpha_{21}^2}{2(1-c)\sigma_3^2} + \frac{\alpha_{22}^2}{2(1-c)\sigma_4^2} - \left(\beta_{21}+\beta_{22}+\mu+\frac{1}{2}\sigma_7^2+\frac{1}{2}\sigma_8^2\right) \\
+ \quad & \frac{2}{c}\frac{\ln k}{k-1} + \frac{2}{c}\frac{\ln k}{k-1} - \sigma_7\frac{W_7(t)}{t} - \sigma_8\frac{W_8(t)}{t}.
\end{aligned}\tag{30}
$$

*By the strong law of large numbers to the Brownian motion, let $k \to \infty$ and then $t \to \infty$, where*

$$\limsup_{t\to\infty} \frac{W_5(t)}{t} = 0, \limsup_{t\to\infty} \frac{W_6(t)}{t} = 0,$$
$$\limsup_{t\to\infty} \frac{W_7(t)}{t} = 0, \limsup_{t\to\infty} \frac{W_7(t)}{t} = 0. \tag{31}$$

*Therefore*

$$\limsup_{t\to\infty} \frac{\ln I_1(t)}{t} \le \frac{\alpha_{11}^2}{2(1-c)\sigma_1^2} + \frac{\alpha_{12}^2}{2(1-c)\sigma_2^2} - \left(\beta_{11} + \beta_{12} + \mu + \frac{1}{2}\sigma_5^2 + \frac{1}{2}\sigma_6^2\right), \tag{32}$$

*and*

$$\limsup_{t\to\infty} \frac{\ln I_2(t)}{t} \le \frac{\alpha_{21}^2}{2(1-c)\sigma_3^2} + \frac{\alpha_{22}^2}{2(1-c)\sigma_4^2} - \left(\beta_{21} + \beta_{22} + \mu + \frac{1}{2}\sigma_7^2 + \frac{1}{2}\sigma_8^2\right). \tag{33}$$

*Finally, let $c \to 0$, $\limsup_{t\to\infty} \frac{\ln I_1(t)}{t}$ and $\limsup_{t\to\infty} \frac{\ln I_2(t)}{t}$ can be obtained as:*

$$\limsup_{t\to\infty} \frac{\ln I_1(t)}{t} \le \frac{\alpha_{11}^2}{2\sigma_1^2} + \frac{\alpha_{12}^2}{2\sigma_2^2} - \left(\beta_{11} + \beta_{12} + \mu + \frac{1}{2}\sigma_5^2 + \frac{1}{2}\sigma_6^2\right), \tag{34}$$

*and*

$$\limsup_{t\to\infty} \frac{\ln I_2(t)}{t} \le \frac{\alpha_{21}^2}{2\sigma_3^2} + \frac{\alpha_{22}^2}{2\sigma_4^2} - \left(\beta_{21} + \beta_{22} + \mu + \frac{1}{2}\sigma_7^2 + \frac{1}{2}\sigma_8^2\right). \tag{35}$$

**Remark 3.1** $G(\sigma_1^2, \sigma_2^2, \sigma_5^2, \sigma_6^2) = \frac{\alpha_{11}^2}{2\sigma_1^2} + \frac{\alpha_{12}^2}{2\sigma_2^2} - \left(\beta_{11} + \beta_{12} + \mu + \frac{1}{2}\sigma_5^2 + \frac{1}{2}\sigma_6^2\right)$ *and*

$G(\sigma_3^2, \sigma_4^2, \sigma_7^2, \sigma_8^2) = \frac{\alpha_{21}^2}{2\sigma_3^2} + \frac{\alpha_{22}^2}{2\sigma_4^2} - \left(\beta_{21} + \beta_{22} + \mu + \frac{1}{2}\sigma_7^2 + \frac{1}{2}\sigma_8^2\right)$ *are decreasing in*
$\sigma_1^2, \sigma_2^2, \sigma_3^2, \sigma_4^2, \sigma_5^2, \sigma_6^2, \sigma_7^2, \sigma_8^2$. *The information will disappearance eventually if*
$\sigma_1^2, \sigma_2^2, \sigma_3^2, \sigma_4^2, \sigma_5^2, \sigma_6^2, \sigma_7^2, \sigma_8^2$ *are large enough, where* $G(\sigma_1^2, \sigma_2^2, \sigma_5^2, \sigma_6^2) < 0$ *and*
$G(\sigma_3^2, \sigma_4^2, \sigma_7^2, \sigma_8^2) < 0$.

## 5 A sufficient condition for the stationary distribution

Theorem 4.1 gives the unique stationary distribution of the existence of stochastic system (5). This also means the stability in a stochastic sense according to [45].

**Theorem 4.1** *The stochastic system* (5) *with initial condition*
$(S_1(0), I_1(0), M_1(0), M_2(0), S_2(0), I_2(0), M_3(0), M_4(0)) \in \mathbb{R}_+^8$ *and the following conditions are satisfied*

$$0 < \Gamma < \min(\xi_1 S_1^2, \xi_2 S_2^2, \xi_3 I_1^2, \xi_4 I_2^2), \tag{36}$$

*where*

$$
\begin{aligned}
\Gamma &= \sigma_1^2 S_1^{*2} I_1^* + \sigma_2^2 S_1^{*2} I_2^{*2} + \sigma_3^2 S_2^{*2} I_1^{*2} + (\sigma_5^2 + \sigma_6^2)\left(\frac{1}{2} I_1^* + 2 I_1^{*2}\right) \\
&\quad + (\sigma_7^2 + \sigma_8^2)\left(\frac{1}{2} I_2^* + 2 I_2^{*2}\right), \\
\xi_1 &= 2\mu - \sigma_1^2 I_1^*, \\
\xi_2 &= 2\mu - \sigma_4^2 I_2^*, \\
\xi_3 &= 2(\beta_{11} + \beta_{12} + \mu - \sigma_5^2 - \sigma_6^2), \\
\xi_4 &= 2(\beta_{21} + \beta_{22} + \mu - \sigma_7^2 - \sigma_8^2),
\end{aligned}
\tag{37}
$$

*then the stationary distribution $\pi$ exists, and the solution of stochastic system (5) is ergodic. Since the mutant population has no impact on the transmission of information, the mutant population is not considered in this section.*

*By the information-existence equilibrium point $E^* = (S_1^*, S_2^*, I_1^*, I_2^*)$, it can be get that*

$$
\lim_{t\to\infty} \frac{1}{t} E \int_0^t \left[\begin{array}{l} \xi_1 (S_1(u) - S_1^*)^2 + \xi_2 (S_2(u) - S_2^*)^2 \\ +\xi_3 (I_1(u) - I_1^*)^2 + \xi_4 (I_2(u) - I_2^*)^2 \end{array}\right] du \leq \Gamma.
\tag{38}
$$

**Proof of Theorem 4.1** *Firstly, define a $\mathcal{C}^2$ function V: by*

$$
\begin{aligned}
\Theta(S_1, S_2, I_1, I_2) &= \Theta_1(I_1) + \Theta_2(I_2) + \Theta_3(S_1, I_1) + \Theta_4(S_2, I_2) \\
&\quad + \Theta_5(S_1, S_2, I_1, I_2),
\end{aligned}
\tag{39}
$$

*where*

$$
\Theta_1(I_1) = I_1 - I_1^* - I_1^* \ln\left(\frac{I_1}{I_1^*}\right), \Theta_2(I_2) = I_2 - I_2^* - I_2^* \ln\left(\frac{I_2}{I_2^*}\right),
$$

$$
\Theta_3(S_1, I_1) = \frac{1}{2}(S_1 + I_1 - S_1^* - I_1^*)^2, \Theta_4(S_2, I_2) = \frac{1}{2}(S_2 + I_2 - S_2^* - I_2^*)^2,
\tag{40}
$$

$$
\Theta_5(S_1, S_2, I_1, I_2) = \frac{1}{2}(S_1 + S_2 + I_1 + I_2 - S_1^* - S_2^* - I_1^* - I_2^*)^2.
$$

*The differential L operator to $\Theta_1$ can be calculated as:*

$$
\begin{aligned}
L\Theta_1 &= (\alpha_{11} S_1 I_1 + \alpha_{12} S_1 I_2 - (\beta_{11} + \beta_{12} + \mu) I_1)\frac{\partial \Theta_1}{\partial I_1} + \frac{1}{2}(\sigma_1^2 S_1^2 I_1^2 + \sigma_2^2 S_1^2 I_2^2 \\
&\quad + \sigma_5^2 I_1^2 + \sigma_6^2 I_1^2)\frac{\partial^2 \Theta_1}{\partial I_1^2} \\
&= (I_1 - I_1^*)\left[\alpha_{11} S_1 + \frac{\alpha_{12} S_1 I_2}{I_1} - (\beta_{11} + \beta_{12} + \mu)\right] + \frac{1}{2}\sigma_1^2 S_1^2 I_1^* + \frac{1}{2}\sigma_2^2 S_1^2 I_2^2 \\
&\quad + \frac{1}{2}\sigma_5^2 I_1^* + \frac{1}{2}\sigma_6^2 I_1^*.
\end{aligned}
\tag{41}
$$

*According to $E^*$, it is easy to get that*

$$\beta_{11} + \beta_{12} + \mu = \alpha_{11} S_1^* + \frac{\alpha_{12} S_1^* I_2^*}{I_1^*}, \tag{42}$$

*and then, $L\Theta_1$ can be expressed as:*

$$
\begin{aligned}
L\Theta_1 &= (I_1 - I_1^*)\left[\alpha_{11}(S_1 - S_1^*) + \alpha_{12}\left(\frac{S_1 I_2}{I_1} - \frac{S_1^* I_2^*}{I_1^*}\right)\right] + \frac{1}{2}\sigma_1^2 S_1^2 I_1^* + \frac{1}{2}\sigma_2^2 S_1^2 I_2^2 \\
&\quad + \frac{1}{2}\sigma_5^2 I_1^* + \frac{1}{2}\sigma_6^2 I_1^* \\
&= (I_1 - I_1^*)\left[\alpha_{11}(S_1 - S_1^*) - \alpha_{12}\frac{S_1^* I_2^* I_1 - S_1 I_2 I_1^*}{I_1 I_1^*}\right] + \frac{1}{2}\sigma_1^2 S_1^2 I_1^* + \frac{1}{2}\sigma_2^2 S_1^2 I_2^2 \\
&\quad + \frac{1}{2}\sigma_5^2 I_1^* + \frac{1}{2}\sigma_6^2 I_1^*.
\end{aligned}
\tag{43}
$$

*By simple calculation, one can get*

$$
\begin{aligned}
L\Theta_1 &\leq \alpha_{11}(S_1 - S_1^*)(I_1 - I_1^*) + \frac{1}{2}\sigma_1^2[(S_1 - S_1^*) + S_1^*]^2 I_1^* \\
&\quad + \frac{1}{2}\sigma_2^2[(S_1 - S_1^*) + S_1^*]^2[(I_2 - I_2^*) + I_2^*]^2 + \frac{1}{2}\sigma_5^2 I_1^* + \frac{1}{2}\sigma_6^2 I_1^*.
\end{aligned}
\tag{44}
$$

*Due to $\frac{1}{2}(x + y)^2 \leq x^2 + y^2$, it is easy to obtain that*

$$
\begin{aligned}
L\Theta_1 &\leq \alpha_{11}(S_1 - S_1^*)(I_1 - I_1^*) + \sigma_1^2(S_1 - S_1^*)^2 I_1^* + \sigma_1^2 S_1^{*2} I_1^* \\
&\quad + \sigma_2^2(S_1 - S_1^*)^2(I_2 - I_2^*)^2 + \sigma_2^2 S_1^{*2} I_2^{*2} + \frac{1}{2}\sigma_5^2 I_1^* + \frac{1}{2}\sigma_6^2 I_1^*.
\end{aligned}
\tag{45}
$$

*Similarly, the differential L operator to $\Theta_2$ can be calculated as:*

$$
\begin{aligned}
L\Theta_2 &= [\alpha_{21} S_2 I_1 + \alpha_{22} S_2 I_2 - (\beta_{21} + \beta_{22} + \mu) I_2]\frac{\partial \Theta_2}{\partial I_2} + \frac{1}{2}(\sigma_3^2 S_2^2 I_1^2 + \sigma_4^2 S_2^2 I_2^2 \\
&\quad + \sigma_7^2 I_2^2 + \sigma_8^2 I_2^2)\frac{\partial^2 \Theta_2}{\partial I_2^2} \\
&= (I_2 - I_2^*)\left[\frac{\alpha_{21} S_2 I_1}{I_2} + \alpha_{22} S_2 - (\beta_{21} + \beta_{22} + \mu)\right] + \frac{1}{2}\sigma_3^2 S_2^2 I_1^2 + \frac{1}{2}\sigma_4^2 S_2^2 I_2^* \\
&\quad + \frac{1}{2}\sigma_7^2 I_2^* + \frac{1}{2}\sigma_8^2 I_2^*.
\end{aligned}
\tag{46}
$$

*According to $E^*$, it is easy to get that*

$$\beta_{21} + \beta_{22} + \mu = \frac{\alpha_{21} S_2^* I_1^*}{I_2^*} + \alpha_{22} S_2^*, \tag{47}$$

*and then, L$\Theta_2$ can be expressed as:*

$$
\begin{aligned}
L\Theta_2 &= (I_2 - I_2^*)\left[\alpha_{22}(S_2 - S_2^*) + \alpha_{21}\left(\frac{S_2 I_1}{I_2} - \frac{S_2^* I_1^*}{I_2^*}\right)\right] + \frac{1}{2}\sigma_3^2 S_2^2 I_1^2 + \frac{1}{2}\sigma_4^2 S_2^2 I_2^* \\
&\quad + \frac{1}{2}\sigma_7^2 I_2^* + \frac{1}{2}\sigma_8^2 I_2^* \\
&= (I_2 - I_2^*)\left[\alpha_{22}(S_2 - S_2^*) - \alpha_{21}\frac{S_2^* I_1^* I_2 - S_2 I_1 I_2^*}{I_2 I_2^*}\right] + \frac{1}{2}\sigma_3^2 S_2^2 I_1^2 + \frac{1}{2}\sigma_4^2 S_2^2 I_2^* \\
&\quad + \frac{1}{2}\sigma_7^2 I_2^* + \frac{1}{2}\sigma_8^2 I_2^*.
\end{aligned}
\tag{48}
$$

*By simple calculation, one can get*

$$
\begin{aligned}
L\Theta_2 &\leq \alpha_{22}(S_2 - S_2^*)(I_2 - I_2^*) + \frac{1}{2}\sigma_3^2[(S_2 - S_2^*) + S_2^*]^2[(I_1 - I_1^*) + I_1^*]^2 \\
&\quad + \frac{1}{2}\sigma_4^2[(S_2 - S_2^*) + S_2^*]^2 I_2^* + \frac{1}{2}\sigma_7^2 I_2^* + \frac{1}{2}\sigma_8^2 I_2^*.
\end{aligned}
\tag{49}
$$

*Due to $\frac{1}{2}(x + y)^2 \leq x^2 + y^2$, it is easy to obtain that*

$$
\begin{aligned}
L\Theta_2 &\leq \alpha_{22}(S_2 - S_2^*)(I_2 - I_2^*) + \sigma_3^2(S_2 - S_2^*)^2(I_1 - I_1^*)^2 + \sigma_3^2 S_2^{*2} I_1^{*2} \\
&\quad + \sigma_4^2(S_2 - S_2^*)^2 I_2^* + \sigma_4^2 S_2^{*2} I_2^* + \frac{1}{2}\sigma_7^2 I_2^* + \frac{1}{2}\sigma_8^2 I_2^*.
\end{aligned}
\tag{50}
$$

*Next, the differential L operator to $\Theta_3$ can be calculated as:*

$$
L\Theta_3 = (S_1 + I_1 - S_1^* - I_1^*)[B_1 - \mu S_1 - (\beta_{11} + \beta_{12} + \mu)I_1] + \frac{1}{2}\sigma_5^2 I_1^2 + \frac{1}{2}\sigma_6^2 I_1^2.
\tag{51}
$$

*According to $E^*$, it is easy to get that*

$$
B = \mu S_1^* + (\beta_{11} + \beta_{12} + \mu)I_1^*,
\tag{52}
$$

*and L$\Theta_3$ can be obtained as:*

$$
\begin{aligned}
L\Theta_3 &= (S_1 + I_1 - S_1^* - I_1^*)[-\mu(S_1 - S_1^*) - (\beta_{11} + \beta_{12} + \mu)(I_1 - I_1^*)] \\
&\quad + \frac{1}{2}(\sigma_5^2 + \sigma_6^2)[(I_1 - I_1^*) + I_1^*]^2.
\end{aligned}
\tag{53}
$$

*By simple calculation, one can get*

$$
\begin{aligned}
L\Theta_3 \leq & -\mu(S_1 - S_1^*)^2 - (\beta_{11} + \beta_{12} + \mu)(S_1 - S_1^*)(I_1 - I_1^*) - \mu(S_1 - S_1^*)(I_1 - I_1^*) \\
& - (\beta_{11} + \beta_{12} + \mu)(I_1 - I_1^*)^2 + (\sigma_5^2 + \sigma_6^2)(I_1 - I_1^*)^2 + (\sigma_5^2 + \sigma_6^2)I_1^{*2} \\
= & -\mu(S_1 - S_1^*)^2 - (\beta_{11} + \beta_{12} + 2\mu)(S_1 - S_1^*)(I_1 - I_1^*) \\
& - (\beta_{11} + \beta_{12} + \mu - \sigma_5^2 - \sigma_6^2)(I_1 - I_1^*)^2 + (\sigma_5^2 + \sigma_6^2)I_1^{*2}.
\end{aligned}
\tag{54}
$$

*Similarly, the differential L operator to $\Theta_4$ can be calculated as*:

$$
\begin{aligned}
L\Theta_4 \leq & -\mu(S_2 - S_2^*)^2 - (\beta_{21} + \beta_{22} + 2\mu)(S_2 - S_2^*)(I_2 - I_2^*) \\
& - (\beta_{21} + \beta_{22} + \mu - \sigma_7^2 - \sigma_8^2)(I_2 - I_2^*)^2 + (\sigma_7^2 + \sigma_8^2)I_2^{*2}.
\end{aligned}
\tag{55}
$$

*Finally, the differential L operator to $\Theta_5$ can be calculated as*:

$$
\begin{aligned}
L\Theta_5 = & (S_1 + S_2 + I_1 + I_2 - S_1^* - S_2^* - I_1^* - I_2^*)[B_1 - \mu S_1 - (\beta_{11} + \beta_{12} + \mu)I_1 \\
& + B_2 - \mu S_2 - (\beta_{21} + \beta_{22} + \mu)I_2] + \frac{1}{2}(\sigma_5^2 + \sigma_6^2)I_1^2 + \frac{1}{2}(\sigma_7^2 + \sigma_8^2)I_2^2 \\
= & (S_1 + S_2 + I_1 + I_2 - S_1^* - S_2^* - I_1^* - I_2^*)[-\mu(S_1 - S_1^*) - \mu(S_2 - S_2^*) \\
& - (\beta_{11} + \beta_{12} + \mu)(I_1 - I_1^*) - (\beta_{21} + \beta_{22} + \mu)(I_2 - I_2^*) \\
& + \frac{1}{2}(\sigma_5^2 + \sigma_6^2)(I_1 - I_1^* + I_1^*)^2 + \frac{1}{2}(\sigma_7^2 + \sigma_8^2)(I_2 - I_2^* + I_2^*)^2 \\
\leq & -\mu(S_1 - S_1^*)^2 - \mu(S_1 - S_1^*)(S_2 - S_2^*) - \mu(S_1 - S_1^*)(I_1 - I_1^*) \\
& - \mu(S_1 - S_1^*)(I_2 - I_2^*) - \mu(S_1 - S_1^*)(S_2 - S_2^*) - \mu(S_2 - S_2^*)^2 \\
& - \mu(S_2 - S_2^*)(I_1 - I_1^*) - \mu(S_2 - S_2^*)(I_2 - I_2^*) - (\beta_{11} + \beta_{12} + \mu)(S_1 \\
& - S_1^*)(I_1 - I_1^*) - (\beta_{11} + \beta_{12} + \mu)(I_1 - I_1^*)^2 - (\beta_{11} + \beta_{12} + \mu)(I_1 - I_1^*)(I_2 \\
& - I_2^*) - (\beta_{21} + \beta_{22} + \mu)(S_1 - S_1^*)(I_2 - I_2^*) - (\beta_{21} + \beta_{22} + \mu)(S_2 - S_2^*)(I_2 \\
& - I_2^*) - (\beta_{21} + \beta_{22} + \mu)(I_1 - I_1^*)(I_2 - I_2^*) - (\beta_{21} + \beta_{22} + \mu)(I_2 - I_2^*)^2 \\
& + (\sigma_5^2 + \sigma_6^2)(I_1 - I_1^*)^2 + (\sigma_5^2 + \sigma_6^2)I_1^{*2} + (\sigma_7^2 + \sigma_8^2)(I_2 - I_2^*)^2 + (\sigma_7^2 + \sigma_8^2)I_2^{*2}.
\end{aligned}
\tag{56}
$$

*Substitute* Eqs ([45]), ([50]), ([54]), ([55]) *and* ([56]), *into* ([39]) *to get*

$$
\begin{aligned}
\Theta(S_1, S_2, I_1, I_2) \quad\leq\quad & \left[\sigma_1^2(S_1 - S_1^*)^2 I_1^* + \sigma_1^2 S_1^{*2} I_1^* + \sigma_2^2 S_1^{*2} I_2^{*2} + \frac{1}{2}\sigma_5^2 I_1^* + \frac{1}{2}\sigma_6^2 I_1^*\right] \\[2mm]
+ \quad & \left[\sigma_4^2(S_2 - S_2^*)^2 I_2^* + \sigma_3^2 S_2^{*2} I_1^{*2} + \sigma_4^2 S_2^{*2} I_2^* + \frac{1}{2}\sigma_7^2 I_2^* + \frac{1}{2}\sigma_8^2 I_2^*\right] \\[2mm]
+ \quad & \left[-\mu(S_1 - S_1^*)^2 - (\beta_{11} + \beta_{12} + \mu - \sigma_5^2 - \sigma_6^2)(I_1 - I_1^*)^2 + (\sigma_5^2 + \sigma_6^2)I_1^{*2}\right] \\[2mm]
+ \quad & \left[-\mu(S_2 - S_2^*)^2 - (\beta_{21} + \beta_{22} + \mu - \sigma_7^2 - \sigma_8^2)(I_2 - I_2^*)^2 + (\sigma_7^2 + \sigma_8^2)I_2^{*2}\right] \\[2mm]
+ \quad & \left[-\mu(S_1 - S_1^*)^2 - \mu(S_2 - S_2^*)^2 - (\beta_{11} + \beta_{12} + \mu - \sigma_5^2 - \sigma_6^2)(I_1 - I_1^*)^2\right. \\[2mm]
- \quad & \left.(\beta_{21} + \beta_{22} + \mu - \sigma_7^2 - \sigma_8^2)(I_2 - I_2^*)^2 + (\sigma_5^2 + \sigma_6^2)I_1^{*2} + (\sigma_7^2 + \sigma_8^2)I_2^{*2}\right] \\[2mm]
= \quad & (\sigma_1^2 I_1^* - 2\mu)(S_1 - S_1^*)^2 + (\sigma_4^2 I_2^* - 2\mu)(S_2 - S_2^*)^2 + [-2(\beta_{11} + \beta_{12} + \mu \\[2mm]
- \quad & \sigma_5^2 - \sigma_6^2)](I_1 - I_1^*)^2 + [-2(\beta_{21} + \beta_{22} + \mu - \sigma_7^2 - \sigma_8^2)](I_2 - I_2^*)^2 \\[2mm]
+ \quad & \sigma_1^2 S_1^{*2} I_1^* + \sigma_2^2 S_1^{*2} I_2^{*2} + \frac{1}{2}\sigma_5^2 I_1^* + \frac{1}{2}\sigma_6^2 I_1^* + \sigma_3^2 S_2^{*2} I_1^{*2} + \sigma_4^2 S_2^{*2} I_2^* \\[2mm]
+ \quad & \frac{1}{2}\sigma_7^2 I_2^* + \frac{1}{2}\sigma_8^2 I_2^* + 2(\sigma_5^2 + \sigma_6^2)I_1^{*2} + 2(\sigma_7^2 + \sigma_8^2)I_2^{*2}.
\end{aligned}
\tag{57}
$$

*By* [Eq (36)], *the ellipsoid*

$$
-\xi_1(S_1 - S_1^*)^2 - \xi_2(S_2 - S_2^*)^2 - \xi_3(I_1 - I_1^*)^2 - \xi_4(I_2 - I_2^*)^2 + \Gamma = 0
\tag{58}
$$

*lies entirely in* $\mathbb{R}_+^8$. *According to* [[45]], *it is easy to know that stochastic system* ([5]) *has a stable stationary distribution.*

**Remark 4.1** *By Theorem 4.1, there exist*

$$
\lim_{(\sigma_1, \sigma_2, \sigma_3, \sigma_4, \sigma_5, \sigma_5, \sigma_7, \sigma_8) \to 0} \Gamma = 0,
$$

$$
\lim_{(\sigma_1, \sigma_2, \sigma_3, \sigma_4, \sigma_5, \sigma_5, \sigma_7, \sigma_8) \to 0} \xi_1 = 2\mu > 0,
$$

$$
\lim_{(\sigma_1, \sigma_2, \sigma_3, \sigma_4, \sigma_5, \sigma_5, \sigma_7, \sigma_8) \to 0} \xi_2 = 2\mu > 0,
\tag{59}
$$

$$
\lim_{(\sigma_1, \sigma_2, \sigma_3, \sigma_4, \sigma_5, \sigma_5, \sigma_7, \sigma_8) \to 0} \xi_3 = 2(\beta_{11} + \beta_{12} + \mu) > 0,
$$

$$
\lim_{(\sigma_1, \sigma_2, \sigma_3, \sigma_4, \sigma_5, \sigma_5, \sigma_7, \sigma_8) \to 0} \xi_4 = 2(\beta_{21} + \beta_{22} + \mu) > 0,
$$

*so that the solution of stochastic system* ([5]) *fluctuates around* $E^*$ *of system* ([1]). *Moreover, the difference between system* ([1]) *and stochastic system* ([5]) *decreases with the values of* $\sigma_1$, $\sigma_2$, $\sigma_3$, $\sigma_4$, $\sigma_5$, $\sigma_5$, $\sigma_7$, $\sigma_8$ *decreasing.*

## 6 The stochastic optimal control model

Based on the stochastic information dissemination model established above, we consider that information cross-fusion plays a positive role in technological innovation and the generation of interdisciplinary knowledge. This paper proposes a control objective to facilitate large-scale information dissemination. Thus, the model's four proportional constants $\alpha_{11}$, $\alpha_{12}$, $\alpha_{22}$, and $\alpha_{21}$ are transformed into control variables $\alpha_{11}(t)$, $\alpha_{12}(t)$, $\alpha_{22}(t)$, and $\alpha_{21}(t)$. These four control variables control the proportion of the populations in the two groups who, respectively, contact the information disseminators of their own group and the other group. Generally, the contact rate can be increased by government policy guidance and organizing information exchange activities.

Hence, the objective function can be proposed as:

$$J(I_1, I_2) = \int_0^{t_f} \left[ I_1(t) + I_2(t) - \frac{c_1}{2}\alpha_{11}^2(t) - \frac{c_2}{2}\alpha_{12}^2(t) - \frac{c_3}{2}\alpha_{21}^2(t) - \frac{c_4}{2}\alpha_{22}^2(t) \right] dt, \tag{60}$$

and the objective function satisfy the state system as:

$$\begin{cases} dS_1(t) &= (B_1 - \alpha_{11}(t)S_1 I_1 - \alpha_{12}(t)S_1 I_2 - \mu S_1)dt \\ &\quad - \sigma_1 S_1 I_1 dW_1(t) - \sigma_2 S_1 I_2 dW_2(t), \\ dS_2(t) &= (B_2 - \alpha_{21}(t)S_2 I_1 - \alpha_{22}(t)S_2 I_2 - \mu S_2)dt \\ &\quad - \sigma_3 S_2 I_1 dW_3(t) - \sigma_4 S_2 I_2 dW_4(t), \\ dI_1(t) &= (\alpha_{11}(t)S_1 I_1 + \alpha_{12}(t)S_1 I_2 - (\beta_{11} + \beta_{12} + \mu)I_1)dt \\ &\quad + \sigma_1 S_1 I_1 dW_1(t) + \sigma_2 S_1 I_2 dW_2(t), \\ dI_2(t) &= (\alpha_{21}(t)S_2 I_1 + \alpha_{22}(t)S_2 I_2 - (\beta_{21} + \beta_{22} + \mu)I_2)dt \\ &\quad + \sigma_3 S_2 I_1 dW_3(t) + \sigma_4 S_2 I_2 dW_4(t). \end{cases} \tag{61}$$

The initial conditions for system (61) are satisfied:

$$S_1(0) = S_{1,0}, S_2(0) = S_{2,0}, I_1(0) = I_{1,0}, I_2(0) = I_{2,0}, \tag{62}$$

where:

$$\alpha_{11}(t), \alpha_{12}(t), \alpha_{21}(t), \alpha_{22}(t) \in U \triangleq \left\{ \begin{array}{l} (\alpha_{11}, \alpha_{12}, \alpha_{21}, \alpha_{22}) | (\alpha_{11}(t), \alpha_{12}(t), \\ \alpha_{21}(t), \alpha_{22}(t)) measurable, \\ 0 \leq \alpha_{11}(t), \alpha_{12}(t), \alpha_{21}(t), \alpha_{22}(t) \leq 1, \\ \forall t \in [0, t_f] \end{array} \right\}, \tag{63}$$

while $U$ is the admissible control set. 0 and $t_f$ are the time interval. The control strength and importance of control measures are expressed as $c_1$, $c_2$, $c_3$ and $c_4$, which are the positive weight coefficients.

**Theorem 5.1** *There exists an optimal control pair* $(\alpha_{11}^*, \alpha_{12}^*, \alpha_{21}^*, \alpha_{22}^*) \in U$, *so that the function is established as*:

$$J(\alpha_{11}^*, \alpha_{12}^*, \alpha_{21}^*, \alpha_{22}^*) = \max\{J(\alpha_{11}, \alpha_{12}, \alpha_{21}, \alpha_{22}) : (\alpha_{11}, \alpha_{12}, \alpha_{21}, \alpha_{22}) \in U\}. \tag{64}$$

**Proof of Theorem 5.1** *Let $X(t) = (S_1(t), S_2(t), I_1(t), I_2(t))^T$ and*

$$
\begin{aligned}
L(t; X(t), \alpha_{11}(t); \alpha_{12}(t); \alpha_{21}(t); \alpha_{22}(t)) \quad &= I_1(t) + I_2(t) - \frac{c_1}{2}\alpha_{11}^2(t) \\
&= \frac{c_2}{2}\alpha_{12}^2(t) - \frac{c_3}{2}\alpha_{21}^2(t) - \frac{c_4}{2}\alpha_{22}^2(t).
\end{aligned}
\tag{65}
$$

*The following five conditions must be satisfied and then the optimal control pair is existence.*

*(i) The set of control variables and state variables is nonempty.*

*(ii) The control set U is convex and closed.*

*(iii) The right-hand side of the state system is bounded by a linear function in the state and control variables.*

*(iv) The integrand of the objective functional is convex on U.*

*(v) There exist constants $d_1$, $d_2 > 0$ and $\rho > 1$ such that the integrand of the objective functional satisfied:*

$$
-L\begin{pmatrix} t; X(t), \alpha_{11}; \\ \alpha_{12}; \alpha_{21}; \alpha_{22} \end{pmatrix} \geq d_1\left(|\alpha_{11}|^2 + |\alpha_{12}|^2 + |\alpha_{21}|^2 + |\alpha_{22}|^2\right)^{\rho/2} - d_2.
\tag{66}
$$

*It is clearly that conditions (i)-(iii) established. Then, the condition (iv) can be easily established such that*

$$
S_1(t) \leq B_1, S_2(t) \leq B_2,
$$

$$
I_1(t) \leq (\alpha_{11}(t)S_1I_1 + \alpha_{12}(t)S_1I_2)dt + \sigma_1 S_1 I_1 dW_1(t) + \sigma_2 S_1 I_2 dW_2(t),
\tag{67}
$$

$$
I_2(t) \leq (\alpha_{21}(t)S_2I_1 + \alpha_{22}(t)S_2I_2)dt + \sigma_3 S_2 I_1 dW_3(t) + \sigma_4 S_2 I_2 dW_4(t).
$$

*Next, for any $t \geq 0$, there is a positive constant M which is satisfied $|X(t)| \leq M$, therefore*

$$
\begin{aligned}
-L\begin{pmatrix} t; X(t), \alpha_{11}; \\ \alpha_{12}; \alpha_{21}; \alpha_{22} \end{pmatrix} &= \frac{c_1\alpha_{11}^2(t) + c_2\alpha_{12}^2(t) + c_3\alpha_{21}^2(t) + c_4\alpha_{22}^2(t)}{2} - I_1(t) - I_2(t) \\
&\geq d_1\left(|\alpha_{11}|^2 + |\alpha_{12}|^2 + |\alpha_{21}|^2 + |\alpha_{22}|^2\right)^{\rho/2} - 2M.
\end{aligned}
\tag{68}
$$

*Let $d_1 = \min\{\frac{c_1}{2}, \frac{c_2}{2}, \frac{c_3}{2}, \frac{c_4}{2}\}$, $d_2 = 2M$ and $\rho = 2$, then condition (v) is established. Hence, the optimal control can be realized.*

**Theorem 5.2** *There exist adjoint variables $\delta_1$, $\delta_2$, $\delta_3$, $\delta_4$ for the optimal control pair*
$(\alpha_{11}^*, \alpha_{12}^*, \alpha_{21}^*, \alpha_{22}^*)$ *that satisfy*:

$$
\begin{cases}
d\delta_1(t) & = \begin{bmatrix} (\delta_1 - \delta_3)(\alpha_{11}(t)I_1 + \alpha_{12}(t)I_2) + \delta_1\mu \\ +\lambda_1(\sigma_1 I_1 + \sigma_2 I_2) - \lambda_3(\sigma_1 I_1 + \sigma_2 I_2) \end{bmatrix} dt - \lambda_1 dW_1 - \lambda_1 dW_2, \\[3mm]
d\delta_2(t) & = \begin{bmatrix} (\delta_2 - \delta_4)(\alpha_{21}(t)I_1 + \alpha_{22}(t)I_2) + \delta_2\mu \\ +\lambda_2(\sigma_3 I_1 + \sigma_4 I_2) - \lambda_4(\sigma_3 I_1 + \sigma_4 I_2) \end{bmatrix} dt - \lambda_2 dW_3 - \lambda_2 dW_4, \\[3mm]
d\delta_3(t) & = 1 + \begin{bmatrix} (\delta_1 - \delta_3)\alpha_{11}(t)S_1 + (\delta_2 - \delta_4)\alpha_{21}(t)S_2 \\ +\delta_3(\beta_{11} + \beta_{12} + \mu) + \lambda_1\sigma_1 S_1 + \lambda_2\sigma_3 S_2 \\ -\lambda_3\sigma_1 S_1 - \lambda_4\sigma_3 S_2 \end{bmatrix} dt + \lambda_3 dW_1 + \lambda_3 dW_2, \\[3mm]
d\delta_4(t) & = 1 + \begin{bmatrix} (\delta_1 - \delta_3)\alpha_{12}(t)S_1 + (\delta_2 - \delta_4)\alpha_{22}(t)S_2 \\ +\delta_4(\beta_{21} + \beta_{22} + \mu) + \lambda_1\sigma_2 S_1 + \lambda_2\sigma_4 S_2 \\ -\lambda_3\sigma_2 S_1 - \lambda_4\sigma_4 S_2 \end{bmatrix} dt + \lambda_4 dW_3 + \lambda_4 dW_4.
\end{cases}
\tag{69}
$$

*With boundary conditions*:

$$
\delta_1(t_f) = \delta_2(t_f) = \delta_3(t_f) = \delta_4(t_f) = 0.
\tag{70}
$$

*In addition, the optimal control pair $(\alpha_{11}^*, \alpha_{12}^*, \alpha_{21}^*, \alpha_{22}^*)$ of state system* (61) *can be given by*:

$$
\begin{aligned}
\alpha_{11}^*(t) &= \min\left\{1, \max\left\{0, \frac{(\delta_1 - \delta_3)S_1 I_1}{c_1}\right\}\right\}, \\[2mm]
\alpha_{12}^*(t) &= \min\left\{1, \max\left\{0, \frac{(\delta_1 - \delta_3)S_1 I_2}{c_2}\right\}\right\}, \\[2mm]
\alpha_{21}^*(t) &= \min\left\{1, \max\left\{0, \frac{(\delta_2 - \delta_4)S_2 I_1}{c_3}\right\}\right\}, \\[2mm]
\alpha_{22}^*(t) &= \min\left\{1, \max\left\{0, \frac{(\delta_2 - \delta_4)S_2 I_2}{c_4}\right\}\right\}.
\end{aligned}
\tag{71}
$$

**Proof of Theorem 5.2** *In order to obtain the expression of optimal control system and optimal control pair, define a Hamiltonian function, which can be written as*:

$$
\begin{aligned}
H &= -I_1(t) - I_2(t) + \frac{c_2}{2}\alpha_{11}^2(t) + \frac{c_2}{2}\alpha_{12}^2(t) + \frac{c_3}{2}\alpha_{21}^2(t) + \frac{c_4}{2}\alpha_{22}^2(t) \\[2mm]
&\quad + \delta_1 \begin{bmatrix} B_1 - \alpha_{11}(t)S_1 I_1 \\ -\alpha_{12}(t)S_1 I_2 - \mu S_1 \end{bmatrix} + \delta_2 \begin{bmatrix} B_2 - \alpha_{21}(t)S_2 I_1 \\ -\alpha_{22}(t)S_2 I_2 - \mu S_2 \end{bmatrix} \\[2mm]
&\quad + \delta_3 \begin{bmatrix} \alpha_{11}(t)S_1 I_1 + \alpha_{12}(t)S_1 I_2 \\ -(\beta_{11} + \beta_{12} + \mu)I_1 \end{bmatrix} + \delta_4 \begin{bmatrix} \alpha_{21}(t)S_2 I_1 + \alpha_{22}(t)S_2 I_2 \\ -(\beta_{21} + \beta_{22} + \mu)I_2 \end{bmatrix} \\[2mm]
&\quad + [-\lambda_1(\sigma_1 S_1 I_1 + \sigma_2 S_1 I_2)] + [-\lambda_2(\sigma_3 S_2 I_1 + \sigma_4 S_2 I_2)] \\[2mm]
&\quad + \lambda_3(\sigma_1 S_1 I_1 + \sigma_2 S_1 I_2) + \lambda_4(\sigma_3 S_2 I_1 + \sigma_4 S_2 I_2).
\end{aligned}
\tag{72}
$$

*According to the Pontyragin maximum principle, the adjoint system can be written as:*

$$\frac{d\delta_1}{dt} = -\frac{\partial H}{\partial S_1}, \frac{d\delta_2}{dt} = -\frac{\partial H}{\partial S_2}, \frac{d\delta_3}{dt} = -\frac{\partial H}{\partial I_1}, \frac{d\delta_4}{dt} = -\frac{\partial H}{\partial I_2},$$

(73)

*and the boundary conditions of adjoint system are*

$$\delta_1(t_f) = \delta_2(t_f) = \delta_3(t_f) = \delta_4(t_f) = 0.$$

(74)

*The optimal control formulae can be written as:*

$$\frac{\partial H}{\partial \alpha_{11}} = c_1 \alpha_{11}(t) - \delta_1 S_1 I_1 + \delta_3 S_1 I_1 = 0,$$

$$\frac{\partial H}{\partial \alpha_{12}} = c_2 \alpha_{12}(t) - \delta_1 S_1 I_2 + \delta_3 S_1 I_2 = 0,$$

$$\frac{\partial H}{\partial \alpha_{21}} = c_3 \alpha_{21}(t) - \delta_2 S_2 I_1 + \delta_4 S_2 I_1 = 0,$$

$$\frac{\partial H}{\partial \alpha_{22}} = c_4 \alpha_{22}(t) - \delta_2 S_2 I_2 + \delta_4 S_2 I_2 = 0.$$

(75)

*And then, the optimal control pair $(\alpha_{11}^*, \alpha_{12}^*, \alpha_{21}^*, \alpha_{22}^*)$ can be calculated based on* Eq (75) *as:*

$$\alpha_{11}^*(t) = \min\left\{1, \max\left\{0, \frac{(\delta_1 - \delta_3)S_1 I_1}{c_1}\right\}\right\},$$

$$\alpha_{12}^*(t) = \min\left\{1, \max\left\{0, \frac{(\delta_1 - \delta_3)S_1 I_2}{c_2}\right\}\right\},$$

$$\alpha_{21}^*(t) = \min\left\{1, \max\left\{0, \frac{(\delta_2 - \delta_4)S_2 I_1}{c_3}\right\}\right\},$$

$$\alpha_{22}^*(t) = \min\left\{1, \max\left\{0, \frac{(\delta_2 - \delta_4)S_2 I_2}{c_4}\right\}\right\}.$$

(76)

**Remark 5.1** *So far, the optimal control system can be got includes state system* (61) *with the initial conditions $S_1(0) = S_{1,0}$, $S_2(0) = S_{2,0}$, $I_1(0) = I_{1,0}$, $I_2(0) = I_{2,0}$ and the adjoint system* (69) *with boundary conditions with the optimization conditions. The optimal control system can be*

*written as*:

$$
\begin{cases}
dS_1(t) &= \left( \begin{array}{l} B_1 - \min\left\{1, \max\left\{0, \dfrac{(\delta_1 - \delta_3)S_1 I_1}{c_1}\right\}\right\} S_1 I_1 \\ -\min\left\{1, \max\left\{0, \dfrac{(\delta_1 - \delta_3)S_1 I_2}{c_2}\right\}\right\} S_1 I_2 - \mu S_1 \end{array} \right) dt \\
&\quad - \sigma_1 S_1 I_1 dW_1(t) - \sigma_2 S_1 I_2 dW_2(t), \\[2ex]
dS_2(t) &= \left( \begin{array}{l} B_2 - \min\left\{1, \max\left\{0, \dfrac{(\delta_2 - \delta_4)S_2 I_1}{c_3}\right\}\right\} S_2 I_1 \\ -\min\left\{1, \max\left\{0, \dfrac{(\delta_2 - \delta_4)S_2 I_2}{c_4}\right\}\right\} S_2 I_2 - \mu S_2 \end{array} \right) dt \\
&\quad - \sigma_3 S_2 I_1 dW_3(t) - \sigma_4 S_2 I_2 dW_4(t), \\[2ex]
dI_1(t) &= \left( \begin{array}{l} \min\left\{1, \max\left\{0, \dfrac{(\delta_1 - \delta_3)S_1 I_1}{c_1}\right\}\right\} S_1 I_1 \\ +\min\left\{1, \max\left\{0, \dfrac{(\delta_1 - \delta_3)S_1 I_2}{c_2}\right\}\right\} S_1 I_2 \\ -(\beta_{11} + \beta_{12} + \mu) I_1 \end{array} \right) dt \\
&\quad + \sigma_1 S_1 I_1 dW_1(t) + \sigma_2 S_1 I_2 dW_2(t), \\[2ex]
dI_2(t) &= \left( \begin{array}{l} \min\{1, \max\{0, \dfrac{(\delta_2 - \delta_4)S_2 I_1}{c_3}\}\} S_2 I_1 \\ +\min\left\{1, \max\left\{0, \dfrac{(\delta_2 - \delta_4)S_2 I_2}{c_4}\right\}\right\} S_2 I_2 \\ -(\beta_{21} + \beta_{22} + \mu) I_2 \end{array} \right) dt \\
&\quad + \sigma_3 S_2 I_1 dW_3(t) + \sigma_4 S_2 I_2 dW_4(t), \\[2ex]
d\delta_1(t) &= \left[ (\delta_1 - \delta_3)\left( \begin{array}{l} \min\left\{1, \max\left\{0, \dfrac{(\delta_1 - \delta_3)S_1 I_1}{c_1}\right\}\right\} I_1 \\ +\min\{1, \max\{0, \dfrac{(\delta_1 - \delta_3)S_1 I_2}{c_2}\}\} I_2 \end{array} \right) + \delta_1 \mu \atop +\lambda_1(\sigma_1 I_1 + \sigma_2 I_2) - \lambda_3(\sigma_1 I_1 + \sigma_2 I_2) \right] dt \\
&\quad - \lambda_1 dW_1 - \lambda_1 dW_2, \\[2ex]
d\delta_2(t) &= \left[ (\delta_2 - \delta_4)\left( \begin{array}{l} \min\left\{1, \max\left\{0, \dfrac{(\delta_2 - \delta_4)S_2 I_1}{c_3}\right\}\right\} I_1 \\ +\min\left\{1, \max\left\{0, \dfrac{(\delta_2 - \delta_4)S_2 I_2}{c_4}\right\}\right\} I_2 \end{array} \right) + \delta_2 \mu \atop +\lambda_2(\sigma_3 I_1 + \sigma_4 I_2) - \lambda_4(\sigma_3 I_1 + \sigma_4 I_2) \right] dt \\
&\quad - \lambda_2 dW_3 - \lambda_2 dW_4, \\[2ex]
d\delta_3(t) &= 1 + \left[ \begin{array}{l} (\delta_1 - \delta_3)\min\left\{1, \max\left\{0, \dfrac{(\delta_1 - \delta_3)S_1 I_1}{c_1}\right\}\right\} S_1 \\ +(\delta_2 - \delta_4)\min\left\{1, \max\left\{0, \dfrac{(\delta_2 - \delta_4)S_2 I_1}{c_3}\right\}\right\} S_2 \\ +\delta_3(\beta_{11} + \beta_{12} + \mu) + \lambda_1 \sigma_1 S_1 + \lambda_2 \sigma_3 S_2 \\ -\lambda_3 \sigma_1 S_1 - \lambda_4 \sigma_3 S_2 \end{array} \right] dt \\
&\quad + \lambda_3 dW_1 + \lambda_3 dW_2, \\[2ex]
d\delta_4(t) &= 1 + \left[ \begin{array}{l} (\delta_1 - \delta_3)\min\left\{1, \max\left\{0, \dfrac{(\delta_1 - \delta_3)S_1 I_2}{c_2}\right\}\right\} S_1 \\ +(\delta_2 - \delta_4)\min\left\{1, \max\left\{0, \dfrac{(\delta_2 - \delta_4)S_2 I_2}{c_4}\right\}\right\} S_2 \\ +\delta_4(\beta_{21} + \beta_{22} + \mu) + \lambda_1 \sigma_2 S_1 + \lambda_2 \sigma_4 S_2 \\ -\lambda_3 \sigma_2 S_1 - \lambda_4 \sigma_4 S_2 \end{array} \right] dt \\
&\quad + \lambda_4 dW_3 + \lambda_4 dW_4,
\end{cases} \tag{77}
$$

*and*

$$\delta_1(t_f) = \delta_2(t_f) = \delta_3(t_f) = \delta_4(t_f) = 0. \qquad (78)$$

## 7 Numerical simulations

The Rung-Kutta algorithm will be used to give some numerical simulations in this section. The results of the numerical simulations show the dissemination characteristics of system (1). Moreover, the results of Theorem 3.1 and Theorem 4.1 of stochastic system (5) will be given by numerical simulations. Due to the range of the parameters has not been explicitly given in previous studies. Therefore, the values of the parameters in the model can be given according to the conditions given by Theorem 3.1 and Theorem 4.1.

The basic reproduction number $R_0$ of system (1) can be easy to get that $R_0 = \frac{B_1 B_2 \alpha_{11} \alpha_{22} - B_1 B_2 \alpha_{12} \alpha_{21}}{\mu^2(\beta_{11} + \beta_{12} + \mu)(\beta_{21} + \beta_{22} + \mu)}$. In order to verify the locally and globally asymptotically stability of information-free equilibrium of system (1). Let $B_1 = 1$, $B_2 = 1$, $\alpha_{11} = 0.003$, $\alpha_{12} = 0.002$, $\alpha_{22} = 0.005$, $\alpha_{21} = 0.003$, $\beta_{11} = 0.002$, $\beta_{12} = 0.004$, $\beta_{22} = 0.003$, $\beta_{21} = 0.005$, $\gamma_{11} = 0.002$, $\gamma_{12} = 0.003$, $\gamma_{22} = 0.003$, $\gamma_{21} = 0.005$, $\mu = 0.1$. It can be concluded that $R_0 = 0.0786 < 1$. And then, Let $B_1 = 1$, $B_2 = 1$, $\alpha_{11} = 0.2$, $\alpha_{12} = 0.003$, $\alpha_{22} = 0.3$, $\alpha_{21} = 0.005$, $\beta_{11} = 0.02$, $\beta_{12} = 0.04$, $\beta_{22} = 0.03$, $\beta_{21} = 0.05$, $\gamma_{11} = 0.02$, $\gamma_{12} = 0.03$, $\gamma_{22} = 0.03$, $\gamma_{21} = 0.05$, $\mu = 0.1$ to verify the locally and globally asymptotically stability of information-existence equilibrium of system (1). It can be concluded that $R_0 = 208.281 > 1$. Figs 2 and 3 verify the stability of information-free and information-existence equilibrium of system (1) respectively, and show that variety groups eventually converge to 0 change over time.

And then we choose the same parameter values in Fig 3 except $\sigma_1$ to $\sigma_8$. In Figs 4 to 7, let $B_1 = 1$, $B_2 = 1$, $\alpha_{11} = 0.2$, $\alpha_{12} = 0.003$, $\alpha_{22} = 0.3$, $\alpha_{21} = 0.005$, $\beta_{11} = 0.02$, $\beta_{12} = 0.04$, $\beta_{22} = 0.03$, $\beta_{21} = 0.05$, $\gamma_{11} = 0.02$, $\gamma_{12} = 0.03$, $\gamma_{22} = 0.03$, $\gamma_{21} = 0.05$, $\mu = 0.1$. Meanwhile, let $\sigma_i(i = 1 \sim 8) = 0.0001$ in Figs 4 and 5. Then, let $\sigma_i(i = 1 \sim 8) = 0.00001$ in Figs 6 and 7. By changing the disturbance intensity to observed the characteristics of stochastic system (5). Figs 4 and 6 confirm the frequency histograms of $S_1(t)$, $S_2(t)$, $I_1(t)$, $I_2(t)$, $M_1(t)$, $M_2(t)$, $M_3(t)$ and $M_4(t)$ respectively under different disturbance intensities. Figs 5 and 7 confirm the comparison between deterministic system (1) and stochastic system (5) of the densities of $S_1(t)$, $S_2(t)$, $I_1(t)$, $I_2(t)$, $M_1(t)$, $M_2(t)$, $M_3(t)$ and $M_4(t)$ respectively change over time under different disturbance intensities. Fig 8 confirms the comparison between $\sigma_i(i = 1 \sim 8) = 0.0001$ and $\sigma_i(i = 1 \sim 8) = 0.00001$ of the densities of (a)$S_1(t)$, (b)$S_2(t)$, (c)$I_1(t)$, (d)$I_2(t)$, (e)$M_1(t)$, (f)$M_2(t)$, (g)$M_3(t)$, (h)$M_4(t)$ change over time.

In Figs 5 and 7, the fluctuating lines represent the population density changes after adding random disturbances. And the stable lines represent the population density changes without adding random disturbances. It can be seen that the population density with added random perturbations is higher than that without added random perturbations. From Figs 5(c), 5(d), 7 (c) and 7(d), the white noise disturbance enhances the population density of information dissemination group $I_1$ and $I_2$. From this, it can be seen that white noise disturbance has a positive effect on information dissemination.

Next, in order to verify the effectiveness of the optimal control strategy, and then observe the change of densities of $I_1(t)$ and $I_2(t)$ when the optimal control strategy is adopted. Here, the optimal control is adopted for the control variables $\alpha_{11}(t)$, $\alpha_{12}(t)$, $\alpha_{22}(t)$, and $\alpha_{21}(t)$, and other parameters remain unchanged. Figs 9 and 10 confirm the densities of $I_1(t)$ and $I_2(t)$ change over time when $\sigma_i(i = 1 \sim 8) = 0.001$ and $\sigma_i(i = 1 \sim 8) = 0.0001$ under constant control measure and optimal control. From Figs 9 and 10, it can be seen that adopting optimal control for

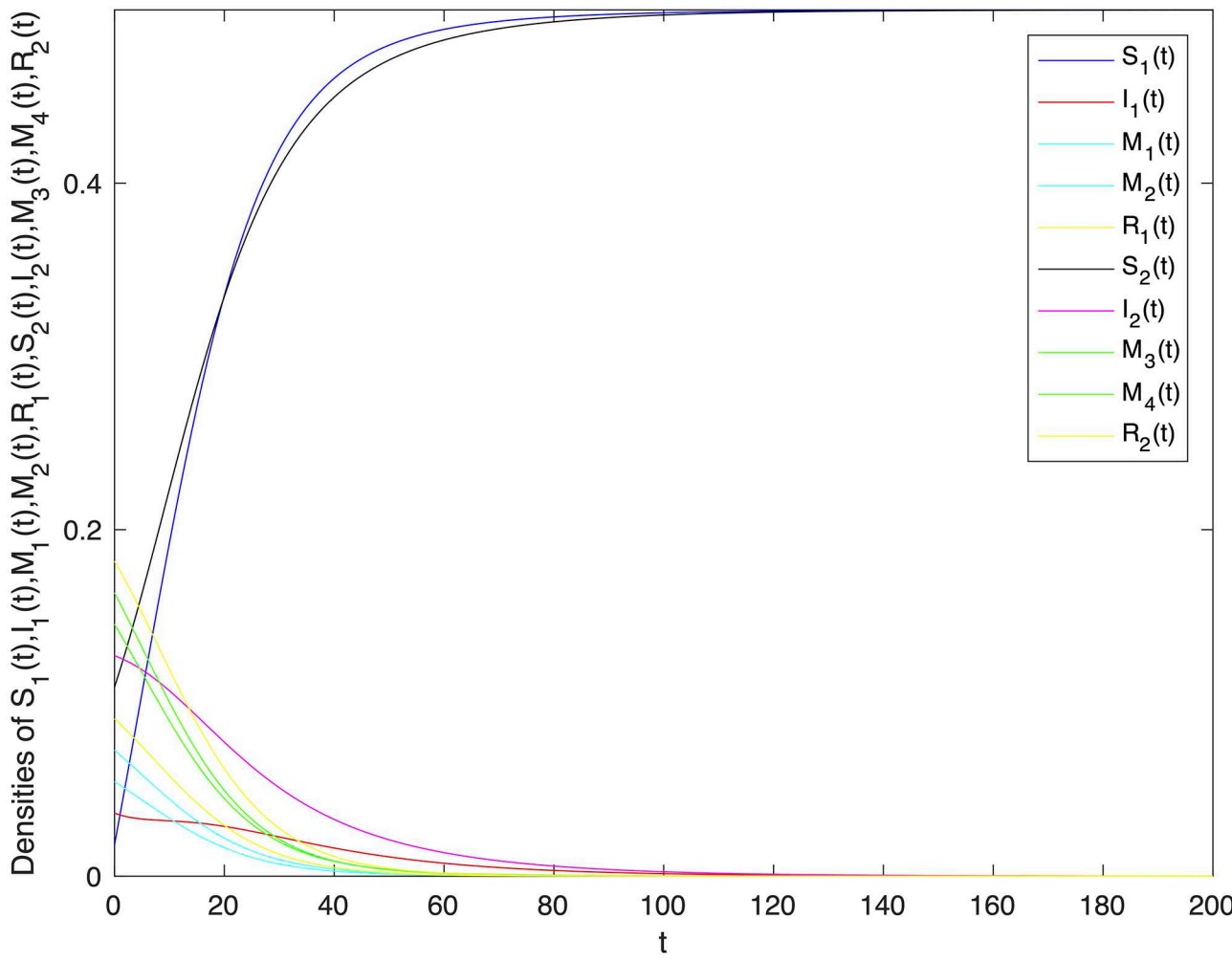

**Fig 2. The stability of information-free equilibrium $E^0$ of system 1 with $R_0 < 1$.**

control parameter $\alpha_{11}(t)$, $\alpha_{12}(t)$, $\alpha_{22}(t)$, and $\alpha_{21}(t)$ can further enhance information dissemination. That is to say, the cross contact rate is more sensitive to the dissemination of information.

Finally, the choice of parameters values has no established principle in the illustrations of the numerical simulations. In relevant literature on information dissemination, the choice of these parameters values does not have a fixed range. Most of them are limited to positive numbers and satisfy the stability condition. In the numerical simulation, the values in other relevant literature are mentioned and the requirements of stability conditions are combined to give the numerical values of the parameters in the model. As for practical problems, determination of the specific numerical parameters is proposed, referring to the relevant professional background knowledge and investigating the actual background with reference to relevant existing literature.

## 8 Conclusion

This study investigates the influence of multi-population information cross-dissemination, mutation, and white noise disturbance on information dissemination. And developed the

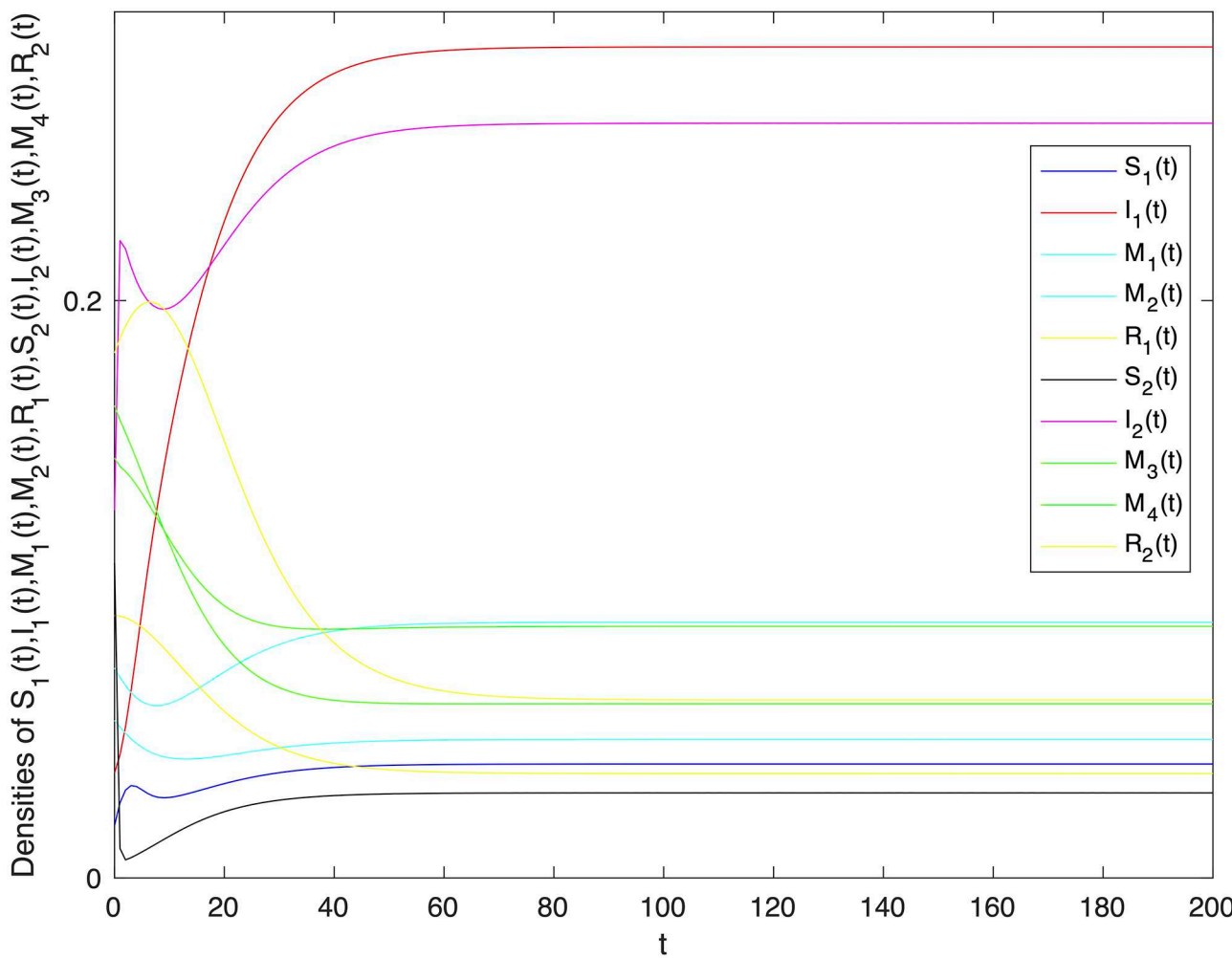

**Fig 3. The stability of information-existence equilibrium $E^*$ of system 1 with $R_0 > 1$.**

stochastic model. The following results are achieved through examination of this paper: (1) White noise disturbance can facilitate the information dissemination, and stochastic environmental factors play a positive role in information dissemination. (2) As disturbance intensity increases, the stochasticity of the model gradually enhances, and the fluctuation of information dissemination trend becomes more apparent. (3) The information dissemination may be effectively facilitated by controlling the perturbation parameters; unlike earlier research, the optimal control strategy provided in this paper is based on the optimal value calculated by the control variables.

To sum up, for positive information, it is necessary to give full play to the activity of the social system itself, and to introduce a large number of stochastic components into the social system to improve the information dissemination. For negative information, on the other hand, it is crucial to eliminate the external environment's uncertain factors and reduce the impact of uncontrollable factors on the social system in order to inhibit its dissemination. In future study, the role of stochastic environmental factors on information dissemination in social systems will be further investigated, and a Lévy process-driven information dissemination model will be developed.

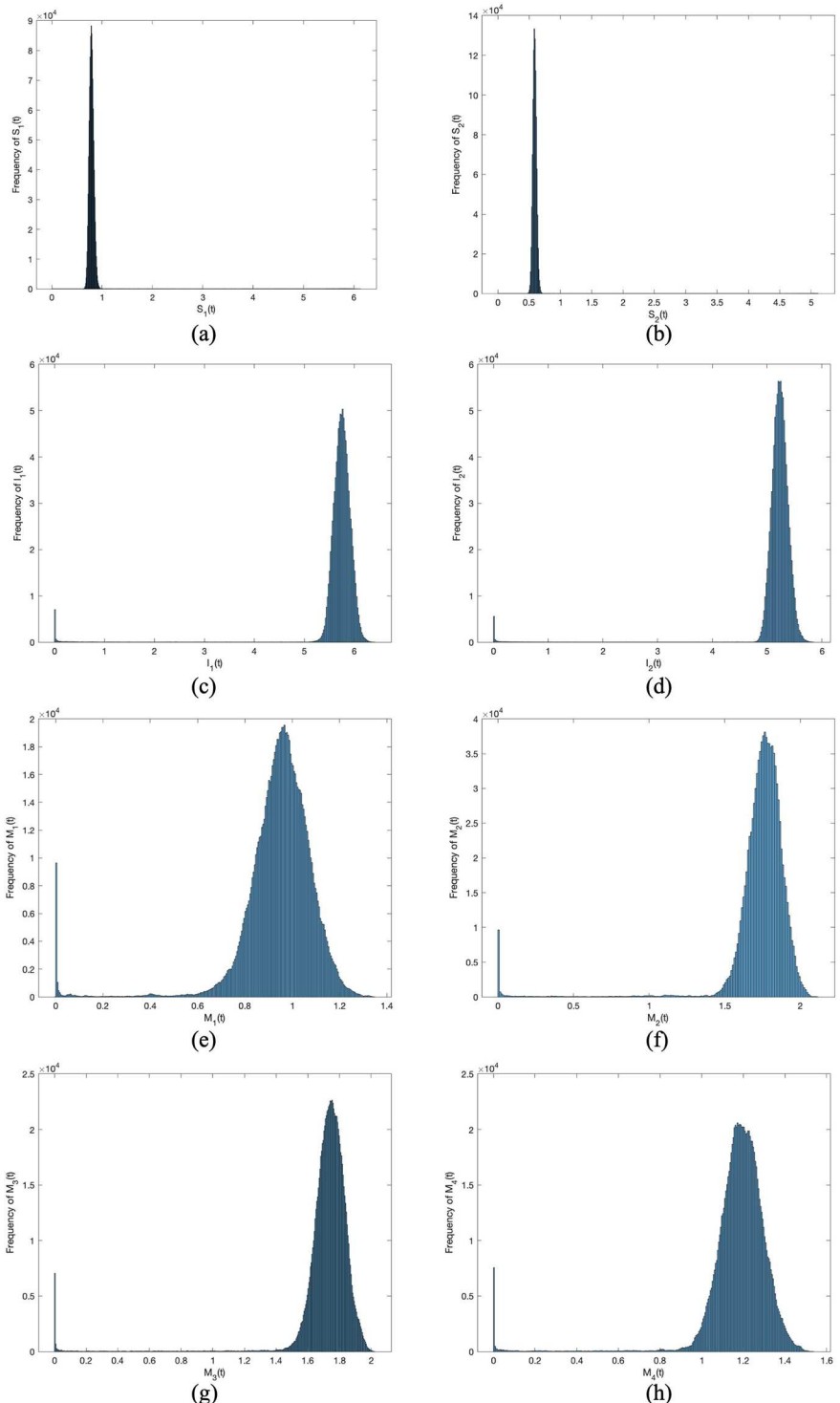

**Fig 4.** Frequency histograms of $(a)S_1(t)$, $(b)S_2(t)$, $(c)I_1(t)$, $(d)I_2(t)$, $(e)M_1(t)$, $(f)M_2(t)$, $(g)M_3(t)$, $(h)M_4(t)$ when $\sigma_i(i = 1 \sim 8) = 0.0001$.

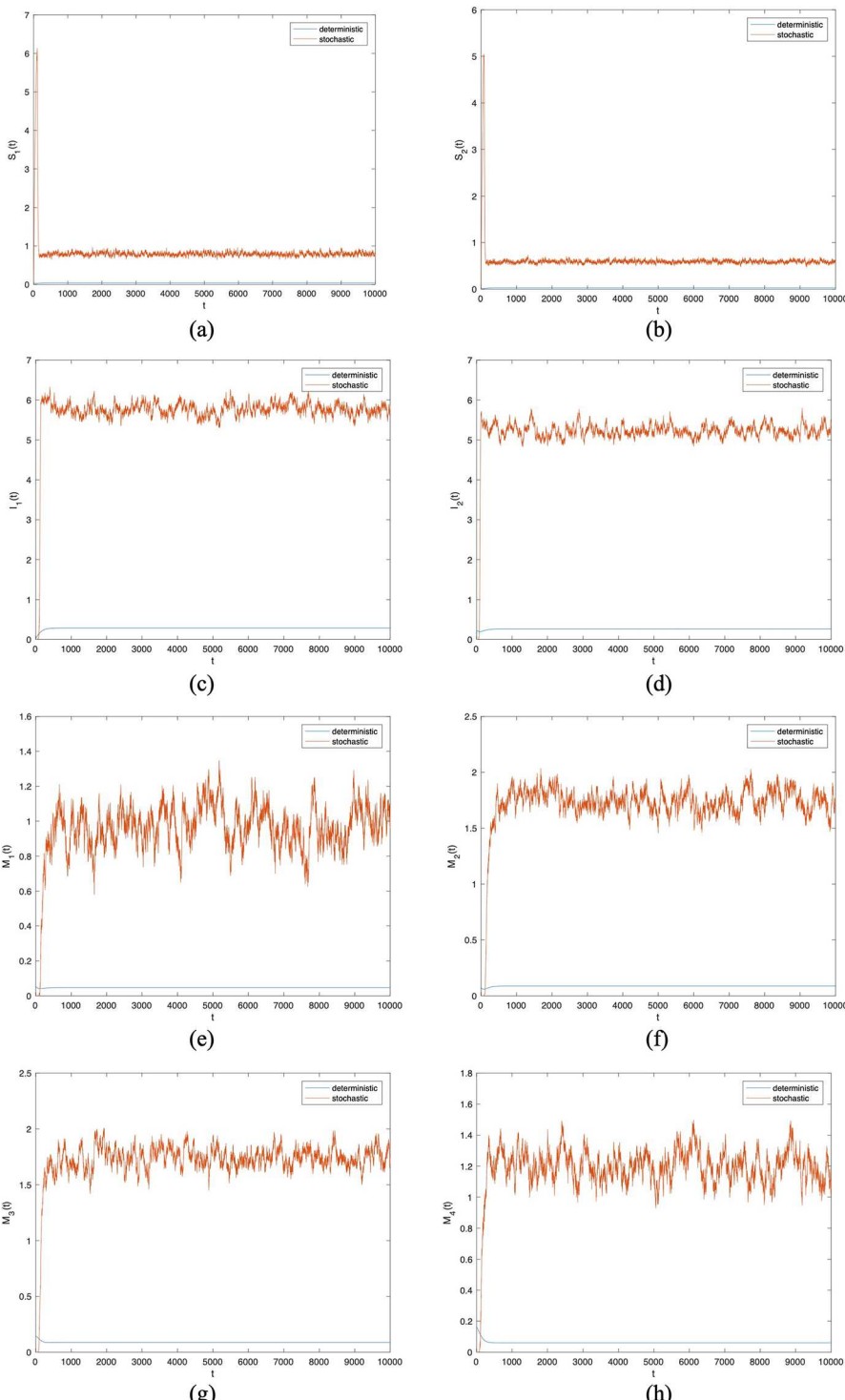

**Fig 5.** Comparison between deterministic model and stochastic model of the densities of $(a)S_1(t)$, $(b)S_2(t)$, $(c)I_1(t)$, $(d)I_2(t)$, $(e)M_1(t)$, $(f)M_2(t)$, $(g)M_3(t)$, $(h)M_4(t)$ change over time when $\sigma_i(i = 1 \sim 8) = 0.0001$.

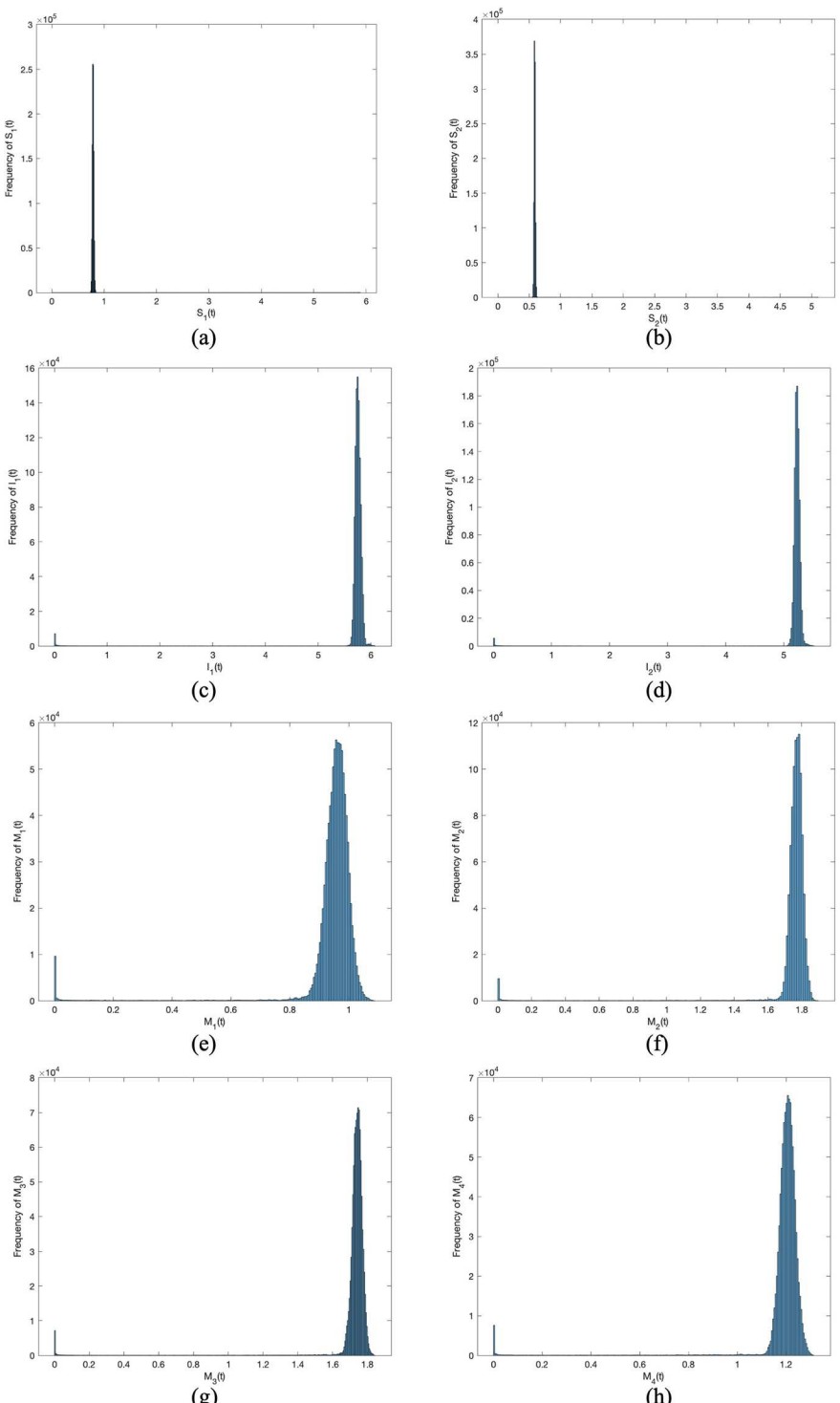

**Fig 6.** Frequency histograms of $(a)S_1(t)$, $(b)S_2(t)$, $(c)I_1(t)$, $(d)I_2(t)$, $(e)M_1(t)$, $(f)M_2(t)$, $(g)M_3(t)$, $(h)M_4(t)$ when $\sigma_i(i = 1 \sim 8) = 0.00001$.

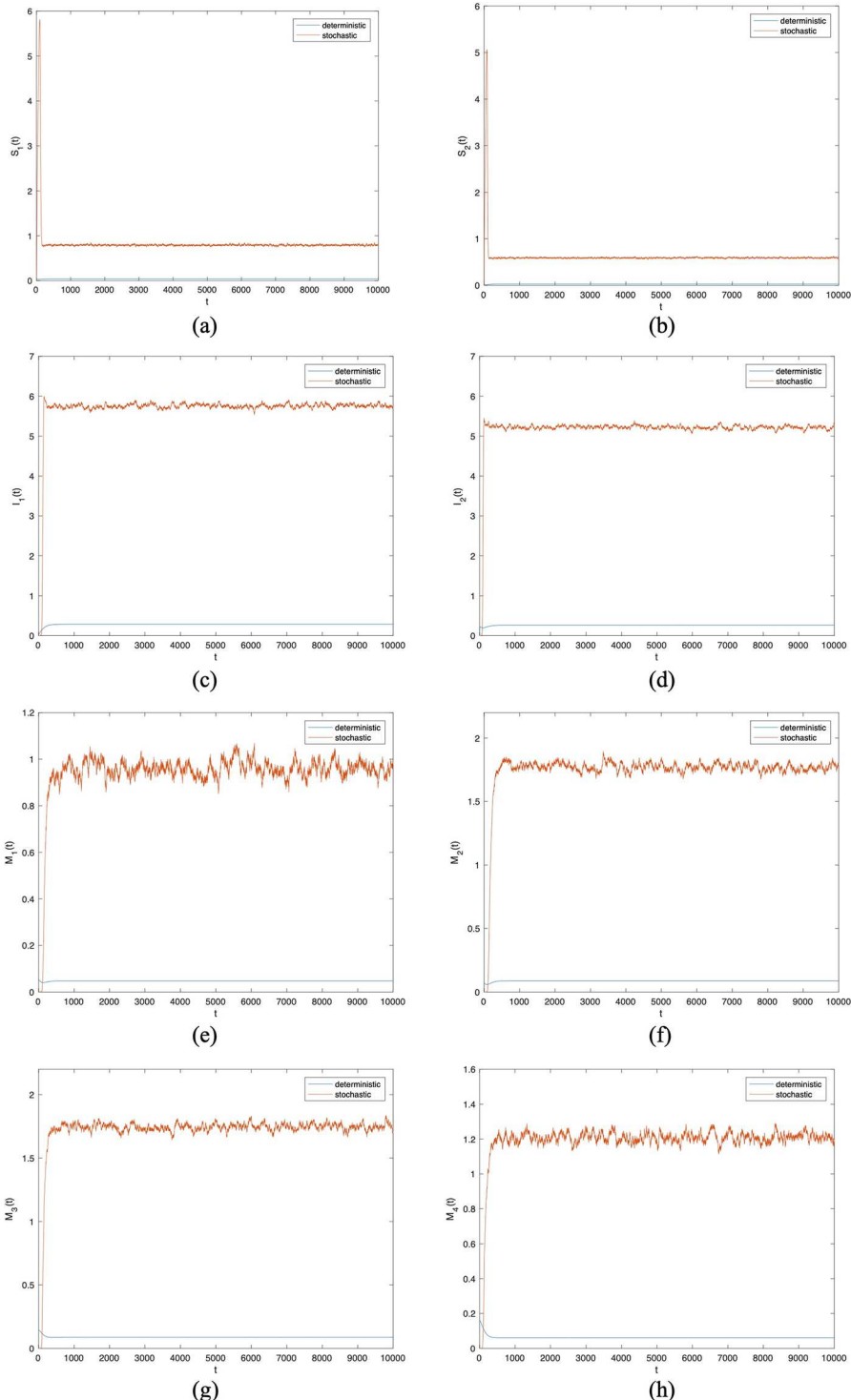

**Fig 7.** Comparison between deterministic model and stochastic model of the densities of $(a)S_1(t)$, $(b)S_2(t)$, $(c)I_1(t)$, $(d)I_2(t)$, $(e)M_1(t)$, $(f)M_2(t)$, $(g)M_3(t)$, $(h)M_4(t)$ change over time when $\sigma_i(i = 1 \sim 8) = 0.00001$.

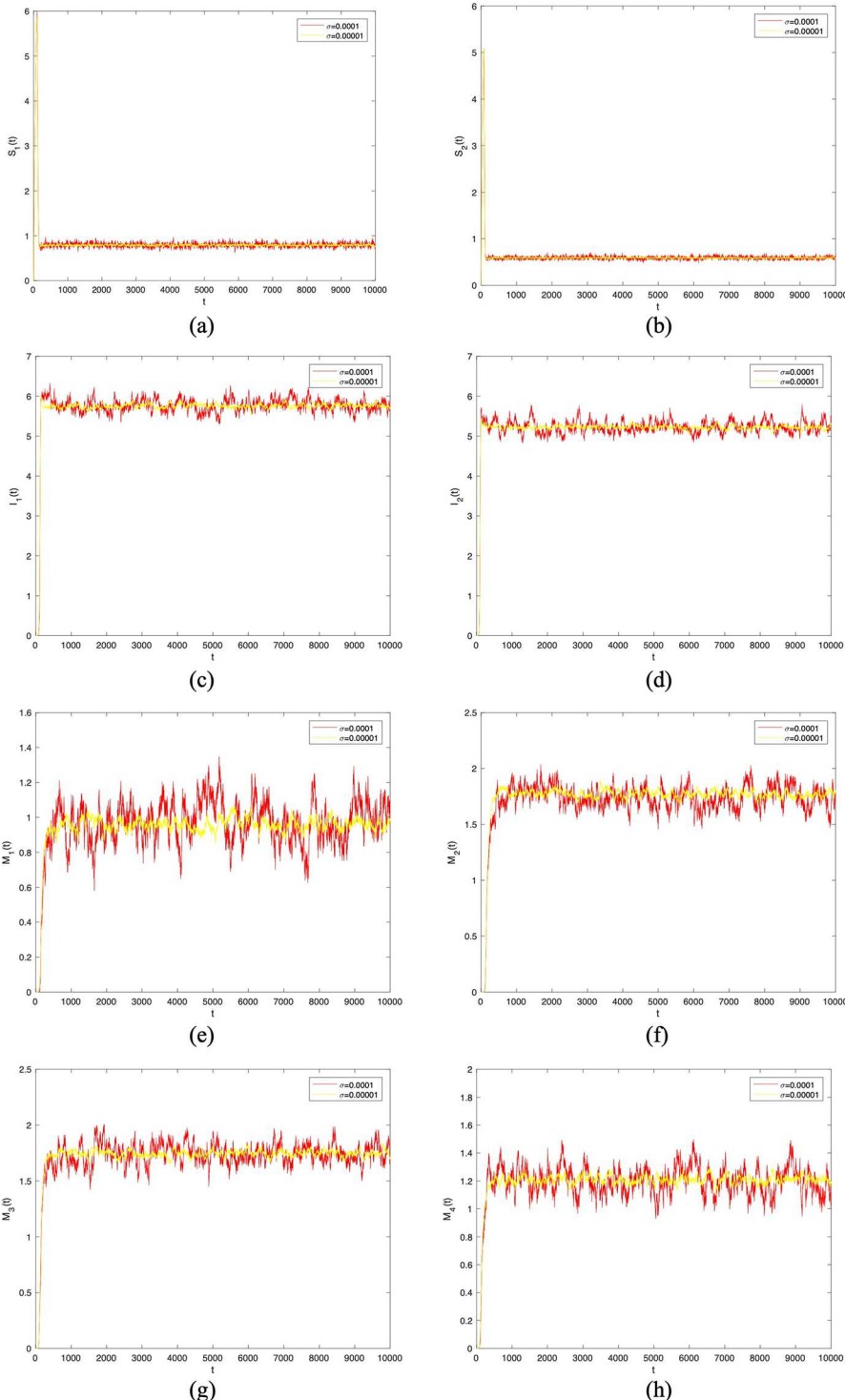

**Fig 8.** Comparison between $\sigma_i(i = 1 \sim 8) = 0.0001$ and $\sigma_i(i = 1 \sim 8) = 0.00001$ of the densities of $(a)S_1(t)$, $(b)S_2(t)$, $(c)$ $I_1(t)$, $(d)I_2(t)$, $(e)M_1(t)$, $(f)M_2(t)$, $(g)M_3(t)$, $(h)M_4(t)$ change over time.

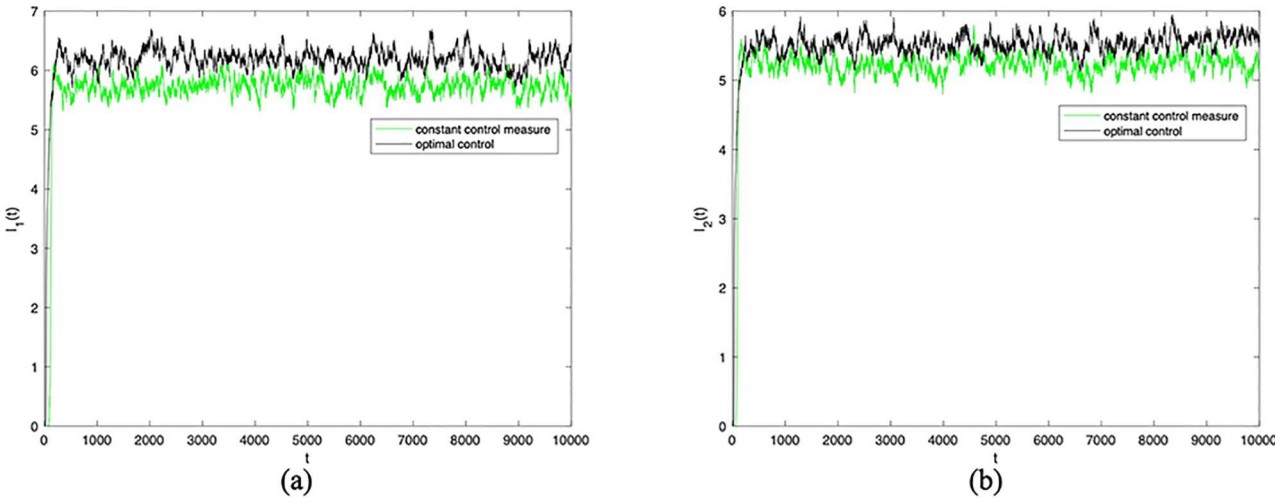

**Fig 9. The densities of $I_1(t)$ and $I_2(t)$ change over time when $\sigma_i(i = 1 \sim 8) = 0.0001$ under constant control measure and optimal control.**

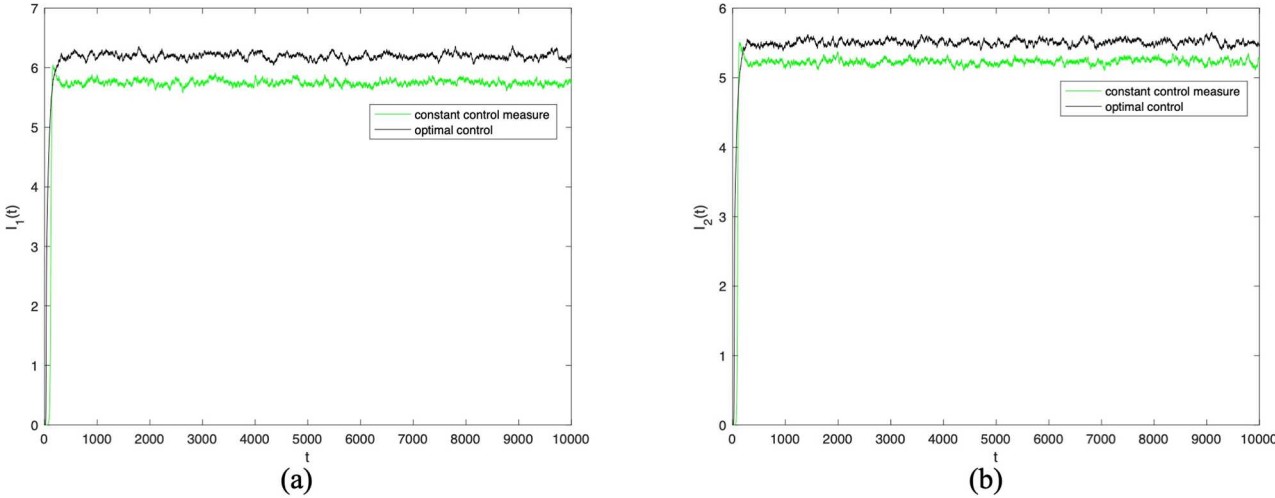

**Fig 10. The densities of $I_1(t)$ and $I_2(t)$ change over time when $\sigma_i(i = 1 \sim 8) = 0.00001$ under constant control measure and optimal control.**

## Author Contributions

**Conceptualization:** Sida Kang, Xilin Hou.

**Data curation:** Sida Kang, Yuhan Hu.

**Formal analysis:** Sida Kang, Tianhao Liu, Yuhan Hu.

**Methodology:** Sida Kang, Hongyu Liu, Xilin Hou.

**Software:** Tianhao Liu, Yuhan Hu.

**Writing – original draft:** Sida Kang.

**Writing – review & editing:** Hongyu Liu, Xilin Hou.

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
