## [Decision Letter · Decision Letter 0]

5 Dec 2023

PONE-D-23-34170Dynamic Analysis and Optimal Control of Stochastic Information Cross-dissemination and Variation Model with Random Parametric PerturbationsPLOS ONE

Dear Dr. Hou,

Thank you for submitting your manuscript to PLOS ONE. After careful consideration, we feel that it has merit but does not fully meet PLOS ONE’s publication criteria as it currently stands. Therefore, we invite you to submit a revised version of the manuscript that addresses the points raised during the review process.

We look forward to receiving your revised manuscript.

Kind regards,

Sanket Kuashik, PhD

Academic Editor

PLOS ONE

Additional Editor Comments:

The author should submit point wise reply to the reviewers comments. I recommend a major revision.

Reviewers' comments:

Reviewer's Responses to Questions

**Comments to the Author**

1. Is the manuscript technically sound, and do the data support the conclusions?

Reviewer #1: Yes

Reviewer #2: Partly

Reviewer #3: Yes

Reviewer #4: Partly

2. Has the statistical analysis been performed appropriately and rigorously? 

Reviewer #1: I Don't Know

Reviewer #2: No

Reviewer #3: N/A

Reviewer #4: N/A

3. Have the authors made all data underlying the findings in their manuscript fully available?

Reviewer #1: Yes

Reviewer #2: No

Reviewer #3: Yes

Reviewer #4: No

4. Is the manuscript presented in an intelligible fashion and written in standard English?

Reviewer #1: Yes

Reviewer #2: No

Reviewer #3: Yes

Reviewer #4: No

5. Review Comments to the Author

Reviewer #1: The reviewer appreciates the paper's presentation of experiments designed by author which seems to contributing the valuable insights to stability inference in a stochastic world model. The reviewer also commends the paper for contributing to the understanding of information dynamics in social systems.

Reviewer #2: Language requires extensive revision. There are many typos and repeated words as in the paragraph between lines 16 and 26. Other examples are shown in the attached PDF file.

Reference 30 cited in the literature review but there is no clear relation with the current study.

Figures 4 to 9 are displayed in a very low resolution and hence many information are not clear.

Reviewer #3: Review Report

Manuscript ID: PONE-D-23-34170

This paper considers that there are many stochastic factors in the social system, which will result in the phenomena of information cross-dissemination and variation. The dual-system stochastic susceptible-infectious-mutant-recovered (2S2I4M2R) model of information cross-dissemination and variation is derived from this problem. Afterward, the existence of the global positive solution is demonstrated, sufficient conditions for the disappearance of information and its stationary distribution are calculated, and the optimal control strategy for the stochastic model is proposed. The numerical simulation supports the results of the theoretical analysis and is compared to the parameter variation of the deterministic model. The results demonstrate that cross-dissemination of information can result in information variation and diffusion. Meanwhile, white noise has a positive effect on information dissemination, which can be improved by adjusting the perturbation parameters.

There is some new contribution and may be consider for publication after addressing the following observations.

1. In abstract the author claims that white noise has a positive effect on information dissemination, which can be improved by adjusting the perturbation parameters.

They may provide the evidence for this either theoretically or numerically. In the current version I did not see any evidence for this claim.

2. The introduction should also make a compelling case for why the study is useful along with a clear statement of its novelty or originality by providing relevant information and providing answers to basic questions such as:

i. What is already known in the literature?

ii. What was done and how it was done?

3. The author may also add some recants work about the current study related to their work in introduction part. They may add the following but not mandatory.

https://doi.org/10.1038/s41598-023-41861-4

https://doi.org/10.1063/1.5016680

https://doi.org/10.1016/j.aej.2023.01.027

doi: 10.3934/math.2023210

4. Even though all the figures are available at the end, but there is no figure in the text and only captions of the figures are there, please see page from 3 onwards.

5. From page 7 onwards there is too many mathematical equations. The author may delete the unnecessary mathematical equations.

6. Improve the quality of the figures (from figure 4 and onwards)

7. In numerical simulation section, what is the criteria to choose the values of the parameters involve in model equation (1) and equation (5).

8. Which parameter is more sensitive and why in numerical simulations section. The author must write some sentences about this.

9. Author may look for some punctuation, typos and editing issues.

10. The conclusion section is too long. In general conclusion consist of what is claimed is achieved.

Reviewer #4: The paper exhibits poor writing quality and contains errors, particularly in the theoretical parts.

Comments:

1) The abbreviation for the suggested model is unclear, and I believe it should be omitted from the text.

2) The introduction lacks consistency and quality in its writing; thus, it requires significant improvement.

3) The construction of the Lyapunov function in Theorem 1 requires revision.

4) The threshold parameter determining the existence of a stationary solution should be based on stochastic perturbations.

6. PLOS authors have the option to publish the peer review history of their article (what does this mean?). If published, this will include your full peer review and any attached files.

Reviewer #1: No

Reviewer #2: No

Reviewer #3: No

Reviewer #4: No

---

## [Author Response · Author response to Decision Letter 0]

8 Apr 2024

The respond to specific reviewer and editor comments are in the file called "Respond to Reviewers".

---

## [Editor Report · Decision Letter 1]

23 Apr 2024

Dynamic Analysis and Optimal Control of Stochastic Information Cross-dissemination and Variation Model with Random Parametric Perturbations

PONE-D-23-34170R1

Dear Author

We’re pleased to inform you that your manuscript has been judged scientifically suitable for publication and will be formally accepted for publication once it meets all outstanding technical requirements.

Kind regards,

Sanket Kaushik, PhD

Academic Editor

PLOS ONE
---

## [Editor Report · Acceptance letter]

10 May 2024

PONE-D-23-34170R1 

PLOS ONE

Dear Dr. Liu, 

I'm pleased to inform you that your manuscript has been deemed suitable for publication in PLOS ONE. Congratulations! Your manuscript is now being handed over to our production team.

Kind regards, 

on behalf of

Dr. Sanket Kaushik 

Academic Editor

PLOS ONE